# ShiftedBronzes: Benchmarking and Analysis of Domain Fine-grained Out-of-Distribution Detection in Gradual Shifts

## Abstract

Out-of-distribution (OOD) detection with fine-grained classes remains a significant challenge due to subtle distribution shifts that can substantially degrade model performance. Existing benchmarks either (i) evaluate multiple, coarse-grained shift levels on general-purpose datasets (e.g., near-OOD & far-OOD), or (ii) examine limited shift level in fine-grained datasets (e.g., open-set recognition on fine-grained dataset), but none comprehensive address multi-level fine-grained shift. To bridge this gap, we introduce ShiftedBronzes, a benchmark designed for fine-grained OOD detection that systematically covers multiple, nuanced shift levels. We evaluate representative post-hoc detectors and vision-language model (VLM)-based methods on ShiftedBronzes and five general OOD benchmarks, uncovering two key limitations: (i) most post-hoc detectors fail to distinguish between varying degrees of distributional shifts, and (ii) although VLMs with prompt tuning outperform most post-hoc methods, they suffer from overfitting when fine-tuned on data with a single background context. To address these challenges, we propose a distribution-aware sensitivity metric that quantifies model robustness across shift levels, and a background-bias mitigation strategy tailored to VLMs. Together, ShiftedBronzes and our sensitivity metric form a comprehensive framework for evaluating and advancing robust multi-level OOD detection in fine-grained domains.

## 1 Introduction

Out-of-distribution (OOD) detection is essential for ensuring the robustness and reliability of computer vision systems in real-world deployments (Yang et al., 2024; Geng et al., 2020), yet most existing studies approaches distributional change with inadequate analytical granularity. Current benchmarks, such as OpenOOD (Yang et al., 2022a) and FS-OOD (Yang et al., 2023), characterize distribution shifts in terms of covariate shift or semantic shift, such as Near-OOD and Far-OOD. In parallel, the open-set recognition (OSR) literature has extended OOD evaluation into the fine-grained setting (Scheirer et al., 2012; Bendale & Boult, 2016) to explore class-level novelty within a narrow semantic space. Some efforts have gone further to consider the hierarchical structure of class labels and its impact on open-set recognition performance (Lang et al., 2024; Verma et al., 2012). However, these studies do not explicitly evaluate model robustness across multiple, progressively distant OOD scenarios.

To address these challenges, we focus on the fine-grained OOD detection setting and leverage cultural-heritage research, where artifact dating serves as a representative proxy task. We therefore construct the ShiftedBronzes benchmark and systematically study OOD robustness under progressively increasing distribution shifts, which instantiates subtle, multi-level shifts for controlled, graded evaluation. Specifically, we use a combination of feature-space visualization and pairwise Bhattacharyya distance (Figure 5) to measure the separability between ID and each OOD group, supporting the ordering of the shift levels by their distributional dissimilarity to the ID domain. Figure 1 illustrates representative examples from our dataset, which includes 2 in-distribution (ID) categories and 5 distinct types of OOD data that progressively differ from the ID data in 4 distribution types.

To quantify the robustness of OOD methods across shift intensities, we propose a novel distribution-aware degree (DAD) metric. This metric captures how sensitively a method responds to varying shift

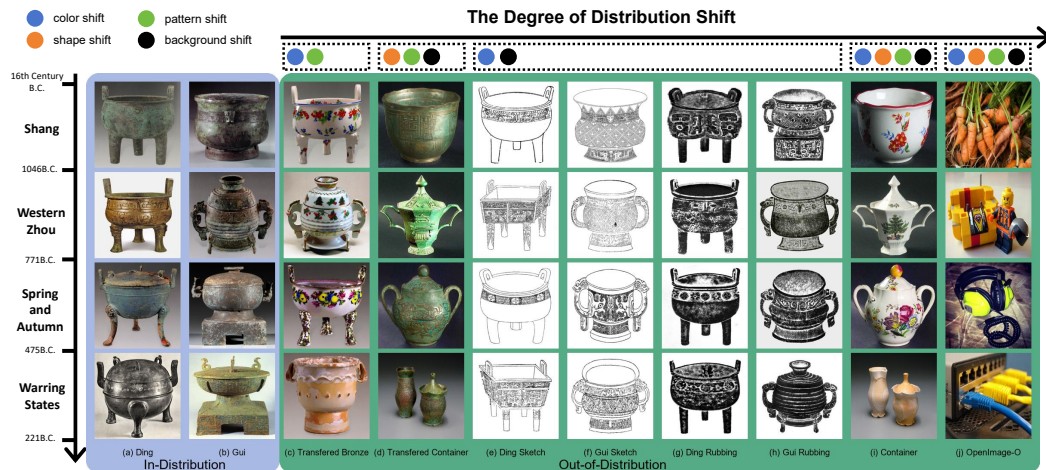

Figure 1: Representative examples of our proposed ShiftedBronzes dataset, including two types ID bronze data and 5 types OOD data with gradual distribution shifts increasing from left to right within the green region. The distribution shifts observed in each type of OOD data are further classified into 4 distinct categories: color shift, pattern shift, shape shift, and background shift.

levels between different OOD dataset and reveals a consistent correlation between shift sensitivity and overall OOD detection performance—offering a new diagnostic perspective.

We conduct a comprehensive evaluation across 6 fine-grained visual classification (FGVC) methods and 16 OOD detection methods, including both post-hoc detectors and pretrained vision-language models (VLMs), on ShiftedBronzes and 5 established general benchmarks. Our results uncover 3 critical limitations of current methods: (i) most post-hoc detectors are insensitive to the degree of distributional shift, leading to degraded OOD detection performance in specialized domains, (ii) detecting covariate shift remains a significant challenge for current state-of-the-art OOD detection methods, and (iii) VLM-based methods, while outperforming most post-hoc techniques overall, are prone to overfitting when fine-tuned on data with homogeneous background context.

Moreover, the OOD shifts captured by ShiftedBronzes are not specific to bronze imagery. Comparable, progressively graded deviations occur in general domain (e.g., stylized distribution shift (Hendrycks et al., 2021a; Wang et al., 2019)) and many specialized domains (e.g., staining or device variation in medical imaging (Cao et al., 2020), seasonal or atmospheric changes in remote sensing (Koßmann et al., 2022), and texture or corrosion in materials inspection (Omee et al., 2024)). The proposed DAD metric is therefore model- and dataset-agnostic, supporting analyses that extend beyond this proxy task and informing benchmark design and method development in other fine-grained settings.

In summary, our contributions include: (i) we introduce ShiftedBronzes, the first benchmark designed to systematically evaluate OOD detection under progressively graded, fine-grained distribution shifts; (ii) we identify key failure modes of existing post-hoc and VLM-based methods, particularly their insensitivity to shift severity and vulnerability to background bias; and (iii) we propose and validate a distribution-aware sensitivity metric that quantifies model robustness across varying shift intensities, offering a new perspective for diagnosing fine-grained OOD performance.

## 2 RELATED WORK

### 2.1 OOD BENCHMARK

A wide range of benchmarks have been developed to evaluate out-of-distribution (OOD) detection methods across general and domain-specific scenarios. OpenOOD (Yang et al., 2022a) provides a unified benchmarking suite that categorizes OOD datasets into multiple difficulty levels (e.g., Near-OOD vs. Far-OOD), supporting standardized evaluation across covariate and semantic shifts. Com-

plementing this, FS-OOD (Yang et al., 2023) proposes a full-spectrum OOD detection paradigm, which redefines the ID-OOD boundary by treating samples exhibiting covariate shifts as part of the ID data, offering a more nuanced categorization of distribution shifts.

Recent studies have extended OOD evaluation to specialized domains such as materials science, drug discovery, and medical imaging. For example, Omee et al. (2024) benchmarked the robustness of graph neural networks (GNNs) for OOD material property prediction; Ji et al. (2022) curated a systematic OOD benchmark for AI-driven drug discovery; and Cao et al. (2020) conducted a comprehensive evaluation of OOD detection techniques across multiple medical imaging modalities. Beyond domain-specific settings, Li et al. (2024); Zhao et al. (2022); Mao et al. (2023) explore distributional robustness by introducing perturbation variables or corruption transformations into general datasets. However, most existing benchmarks fall short in systematic evaluating of model robustness across multi-level fine-grained distribution shifts prevalent in specialized domains.

### 2.2 OOD Detection Methods

The field of OOD detection has seen rapid progress, with two dominant lines of work: post-hoc methods and vision-language model (VLM)-based methods. Post-hoc methods operate on pretrained models without architectural modifications and focus on scoring mechanisms to differentiate ID from OOD inputs. Hendrycks & Gimpel (2017) introduced the maximum softmax probability (MSP) as a simple baseline. Subsequent work has proposed more advanced indicators, including Mahalanobis distance (Lee et al., 2018; Liu & Qin, 2024), nearest neighbors (Park et al., 2023; Sun et al., 2022), energy-based scores (Liu et al., 2020), gradient sensitivity (Huang et al., 2021a; Liang et al., 2018), Gram matrix statistics (Sastry & Oore, 2020), rectified activations (Xu et al., 2024; Djurisic et al., 2023) and sparsity-inducing mechanisms (Sun & Li, 2022).

With the advent of vision-language models like CLIP (Radford et al., 2021), recent approaches leverage their multi-modal representations for OOD detection. For zero-shot OOD detection methods, Ming et al. (2022) and Miyai et al. (2025) separated ID and OOD by aligning visual features with textual concepts. CLIPN (Wang et al., 2023) introduced a negative prompt generation mechanism for scoring non-ID concepts. Furthermore, prompt learning methods such as CoOp (Zhou et al., 2022) have shown effectiveness and efficiency in few-shot ID classification. LoCoOp (Miyai et al., 2024) utilized local CLIP features to regularize representations with OOD-aware priors during training. ID-like (Bai et al., 2024) proposed a prompt-learning strategy that identifies OOD samples semantically close to ID data, refining discrimination boundary. While effective, prompt learning methods are often sensitive to contextual bias and background homogeneity, particularly in fine-grained tasks.

### 2.3 Bronze Dating

We adopt ancient Chinese bronze dating as a proxy to evaluate model robustness across multiple, progressively distant OOD scenarios. Bronze dating involves identifying the historical period of ancient Chinese bronze. Early work primarily focused on physical or chemical analysis (Doménech-Carbó et al., 2014; 2018; Wang et al., 2021; Ling et al., 2007), such as corrosion layer composition or metallurgical signatures. Recent efforts have formulated the problem as a fine-grained visual classification (FGVC) task. Zhou et al. (2023) constructed a dataset of bronze Ding and benchmarked several FGVC methods (Yang et al., 2022b; Chang et al., 2021; Chen et al., 2022; Yang et al., 2018; Huang et al., 2021b), proposing a multi-granularity model to capture structural cues across eras. The naturally ordered, fine-grained OOD shifts in bronze dating offer a principled setting to assess model robustness across progressively distant OOD scenarios.

## 3 ShiftedBronzes Benchmark

We introduce a comprehensive benchmark ShiftedBronzes tailored for multi-level fine-grained OOD detection. ShiftedBronzes includes 2 ID categories and 5 distinct types of OOD data that progressively differ from the ID data in 4 distribution types.

### 3.1 Bronze, Sketch and Rubbing Data

**Data Collection.** We expand the bronze Ding dataset (Zhou et al., 2023) by collecting 2518 bronze Gui images from both five published archaeology books and four websites, which are organized into color, sketch, and rubbing categories. The statistics of Ding and Gui dataset are presented in Table 1. Color images of bronzes were employed as ID data for training the bronze dating model. Sketch and rubbing images fall within the ID domain under the FS-OOD (Yang et al., 2023) setting, which exhibit covariate shift relative to the ID images. However, considering the practical need to detect covariate shifts in real-world scenarios, we treat sketch and rubbing images as OOD in our benchmark.

| Era | Shang | | Western Zhou | | | Spring and Autumn | | | Warring States | | | Total |
|---|---|---|---|---|---|---|---|---|---|---|---|---|
| | Early | Late | Early | Mid | Late | Early | Mid | Late | Early | Mid | Late | |
| Bronze Data (ID) | | | | | | | | | | | | |
| Ding | 93 | 945 | 801 | 410 | 264 | 298 | 189 | 215 | 85 | 78 | 141 | 3519 |
| Gui | 8 | 355 | 732 | 550 | 352 | 138 | 16 | 27 | 8 | 5 | 1 | 2192 |
| Sketch and Rubbing Data (OOD) | | | | | | | | | | | | |
| Ding | 0 | 67 | 51 | 18 | 15 | 4 | 4 | 2 | 2 | 0 | 8 | 171 |
| Gui | 0 | 46 | 113 | 72 | 78 | 14 | 2 | 0 | 0 | 0 | 1 | 326 |
| Transferred Container Data (OOD) | | | | | | | | | | | | |
| | 918 | 11718 | 13805 | 8166 | 5544 | 3924 | 1854 | 2196 | 846 | 756 | 1296 | 51023 |

Table 1: Statistics of bronze, sketch, rubbing and transferred container data.

**Data Annotation.** All bronze Gui have been annotated with comprehensive expert knowledge, including era (4 course-grained dynasties and 11 fine-grained periods), attributes (35 shapes and 149 characteristics with bounding boxes), literature, locations of excavation, and current exhibition museums. The statistics of shapes and characteristics labels are shown in Figure 2 (a) and (b). Details of data collection, data annotation and category information are provided in the appendix.

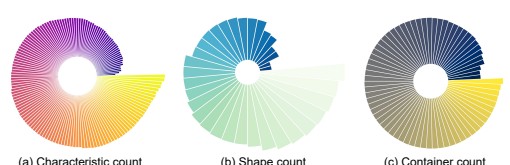

Figure 2: Statistics of (a) characteristic annotation, (b) shape annotation, (c) container categories.

## 3.2 CONTAINER DATA

**Data Collection.** As shown in Figure 3, the process of collecting container data can be divided into three steps: **Step 1**: We employed an open-source LLM (Moonshot AI Technology Co., 2024) to sift through the 21,841 categories of ImageNet-21K, selecting those that are related to the concept of containers, such as "bowl", "cup". A total of 94 categories related to the concept of containers were identified, encompassing 76399 images; **Step 2**: We removed those categories that appeared in the

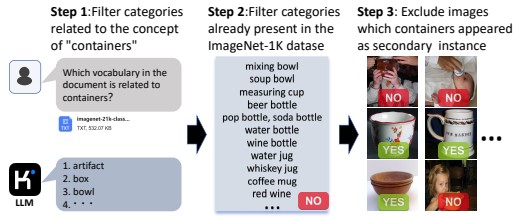

Figure 3: The detailed process for collecting and filtering container data.

ImageNet-1K dataset, leaving 83 categories; **Step 3**: We removed images in which the containers are secondary instances, such as a person holding a wine glass, leaving 51,023 container images as our container data. The statistics of the container data is illustrated in Figure 2 (c).

## 3.3 TRANSFERRED OOD DATA

We leverage a zero-shot material transfer model, ZeST (Cheng et al., 2025), to operate between bronze and container images, generating two categories of transferred images: (1) **Transferred Container Images**, which maintain the color and background of bronzes while exhibiting container shape and distorted bronze patterns. By transferring the materials of bronzes onto containers, we can obtain 51,023 transferred container data that combine the material characteristics of bronzes with the shapes of modern containers, as shown in Figure 4 (a). To prevent information leakage from the training data, we used 2861 bronze images from the test set to perform material transfers on

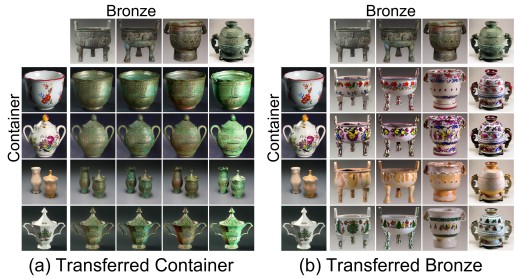

Figure 4: The material transferring process between container and bronze images.

| Method | OA | Bronze Category (Ding/Gui) | | | | | | | | | | |
| | | Shang | | Western Zhou | | | Spring and Autumn | | | Warring States | | |
| | | Early | Late | Early | Mid | Late | Early | Mid | Late | Early | Mid | Late |
| **Single-Granularity** | | | | | | | | | | | | |
| NTS-Net (Yang et al., 2018) | 73.19 | 82.35 | 73.58 | 79.27 | 74.58 | **79.55** | 61.47 | 55.34 | 71.31 | 30.23 | 35.9 | 71.83 |
| SPS (Huang et al., 2021b) | 75.74 | 82.35 | 79.72 | 79.27 | **76.04** | 76.3 | 70.18 | 58.25 | 68.85 | 51.16 | 35.9 | 83.1 |
| P2PNet (Yang et al., 2022b) | 76.03 | **88.24** | 70.05 | **84.75** | 75.42 | 75 | **81.65** | 53.4 | **75.41** | 48.84 | **41.03** | **88.73** |
| **Multi-Granularity** | | | | | | | | | | | | |
| YourFL (Chang et al., 2021) | 99.54 | 99.60/99.45 | | | | | | | | | | |
| | 82.37 | 72.65 | | 88.17 | | | 79.01 | | | 77.78 | | |
| | 70.28 | 68.63 | 73.12 | 76.4 | 70.62 | 68.51 | 68.35 | 52.43 | 62.3 | 27.91 | **41.03** | 71.83 |
| HRN (Chen et al., 2022) | 39.57 | 25.24/62.75 | | | | | | | | | | |
| | 85.98 | 78.21 | | 90.68 | | | 82.62 | | | 83.66 | | |
| | 73.85 | 78.43 | 75.42 | 80.18 | 75.21 | 70.13 | 72.02 | 54.37 | 71.31 | 39.53 | 23.08 | 81.69 |
| AKG (Zhou et al., 2023) | 99.72 | 99.66/99.82 | | | | | | | | | | |
| | 87.84 | 82.76 | | 91.51 | | | 85.78 | | | 80.12 | | |
| | 77.88 | **88.24** | **81.11** | 82.4 | 75.42 | **79.55** | 74.77 | **63.11** | 70.49 | **53.49** | 33.33 | 84.51 |

Table 2: The comparison of six FGVC methods on the bronze dataset. We report the overall accuracy and individual accuracy for different individual era and bronze category. Bold indicates the best performance within each granularity level.

51023 container images, with an average of 17 container images corresponding to each bronze image. Consequently, the transferred container data can also be correlated with each era based on the corresponding bronze images. The statistics for the transferred container data are presented in Table 1. (2) **Transferred Bronze Images**, which preserve the color and shape of bronzes while exhibiting container patterns and background. by transferring the materials of modern containers onto bronzes, we can obtain 51,023 transferred bronze data that combine the material characteristics of containers with the shapes of bronzes, as illustrated in Figure 4 (b). We used 51023 container images to perform material transfers on the 2861 bronze test data. The statistics of transferred bronze data are consistent with those of the container data shown in Figure 2 (b).

### 3.4 DISTRIBUTION SHIFT ANALYSIS

We conduct a comprehensive analysis of the distribution shifts present in various data types within the ShiftedBronzes from both qualitative and quantitative perspectives. Initially, we randomly selected 200 images from each data type in the ShiftedBronzes dataset, as well as from 5 widely-used OOD datasets. We then employed the pre-trained ResNet-50 He et al. (2016) and ViT-B/16 Dosovitskiy et al. (2021) models, trained on ImageNet-1K, to extract image features. To facilitate visualization, we applied PCA to reduce the dimensionality of the extracted features and generated scatter plots (see Figure 5 (a)). The results demonstrate that the OOD data types in ShiftedBronzes exhibit varying degrees of distribution shift when compared with the bronze data.

In addition, we computed the Bhattacharyya Distance (BD) between the reduced dimensional featuress of bronze images and those of various OOD datasets to quantify the distribu-

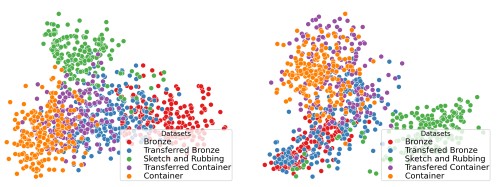

(a) Feature visualization of ShiftedBronzes.

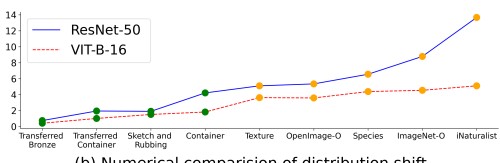

(b) Numerical comparision of distribution shift.

Figure 5: (a) Feature visualization of data in ShiftedBronzes extracted using ResNet-50 (left) and ViT-B-16 (right) models. (b) The BD values between the features of ID images and those of various OOD datasets.

tion shift relative to the bronze data. The BD values were plotted as line charts (see Figure 5(b)). The BD values from ResNet-50 show an increasing trend across our data variants. In contrast, with features from ViT-B/16, we observe that the BD of Transferred Container and Sketch and Rubbing to the ID data are nearly indistinguishable. Taking into account the results from both ResNet-50 and ViT-B/16, we define the degree of distribution shift in the following order: Transferred Bronze → Transferred Container → Sketch and Rubbing → Container → Other General OOD Data.

## 4 EXPERIMENTS

### 4.1 DATA PREPARATION

The bronze data from our proposed ShiftedBronzes is used as ID data to train the bronze dating methods. Following the divisions of bronze Ding dataset (Zhou et al., 2023), we split the 5711 bronze Ding and Gui images into three sets: the scales of the training set, validation set, and test set

| Method | Type | Transferred Container FPR95↓ AUROC↑ | Sketch and Rubbing FPR95↓ AUROC↑ | Transferred Bronze FPR95↓ AUROC↑ | Container FPR95↓ AUROC↑ | Average FPR95↓ AUROC↑ |
|---|---|---|---|---|---|---|
| DICE (Sun & Li, 2022) | Post-hoc | 94.15/38.35 | 98.39/26.74 | 75.11/67.79 | 88.68/52.48 | 89.08/46.34 |
| EBO (Liu et al., 2020) | | 61.41/80.54 | 87.27/58.15 | 70.51/75.45 | 64.86/78.95 | 71.01/73.27 |
| GradNorm (Huang et al., 2021a) | | 65.27/79.56 | 89.16/57.53 | 64.02/77.46 | 53.92/84.05 | 68.09/74.65 |
| Gram (Sastry & Oore, 2020) | | 87.20/61.75 | 83.83/28.37 | 95.24/50.25 | 78.52/68.92 | 86.2/52.32 |
| KL-Matching (Basart et al., 2022) | | 96.17/52.77 | 95.88/43.43 | 95.47/60.14 | 97.43/50.18 | 96.24/51.63 |
| KNN (Sun et al., 2022) | | 45.40/86.94 | 89.26/58.54 | 60.06/81.07 | 40.93/89.39 | 58.91/78.98 |
| MDS (Lee et al., 2018) | | **22.44**/92.20 | 93.73/57.11 | 36.72/87.03 | 14.95/95.17 | 41.96/82.88 |
| MLS (Basart et al., 2022) | | 61.48/80.14 | 87.27/58.24 | 70.51/75.29 | 65.02/78.66 | 71.07/73.08 |
| MSP (Hendrycks & Gimpel, 2017) | | 68.55/71.57 | 88.78/60.35 | 72.15/71.39 | 68.55/72.17 | 74.51/68.87 |
| ODIN (Liang et al., 2018) | | 77.52/81.14 | 55.34/85.02 | 80.87/72.90 | 88.52/82.01 | 75.56/80.27 |
| OpenMax (Bendale & Boult, 2016) | | 70.00/72.64 | 89.84/62.33 | 73.41/69.93 | 70.03/69.85 | 75.82/68.69 |
| ReAct (Sun et al., 2021) | | 63.76/78.80 | 87.30/54.94 | 64.89/77.34 | 65.18/78.66 | 70.28/72.44 |
| VIM (Wang et al., 2022) | | 22.77/94.89 | 86.43/55.05 | 38.78/90.60 | 17.94/96.30 | 41.48/84.21 |
| CLIPN (Wang et al., 2023) | VLM-based (origin bg) | 69.65/69.34 | 58.55/82.87 | 48.82/80.61 | 7.75/97.72 | 46.19/82.64 |
| LoCoOp (Miyai et al., 2024) | | 30.72/92.25 | **8.45**/**98.49** | 7.09/98.63 | 10.39/97.02 | 16.41/95.9 |
| ID-like (Bai et al., 2024) | | 22.72/**94.43** | 24.55/94.94 | **5.13**/**98.8** | **0.55**/**99.79** | **13.24**/**96.99** |

Table 3: The comparison of sixteen OOD detection methods on the ShiftedBronzes. We categorized the comparative methods into two types based on their mechanisms and reported their FPR95 and AUPOC. Bold indicates the best performance, while underlined denotes the second-best performance.

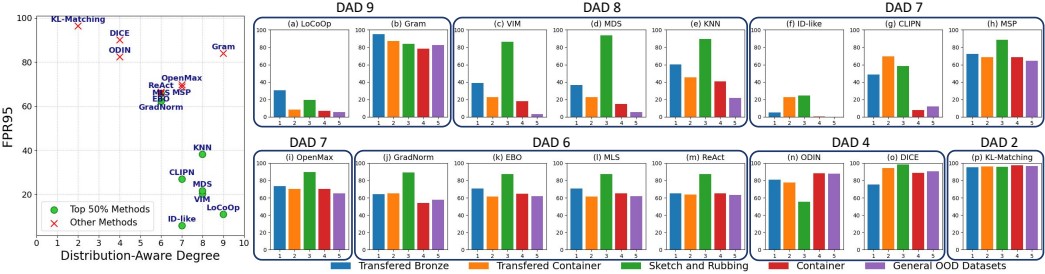

Figure 6: The comparison of FPR95↓ performance of 16 OOD detection methods on the OOD data in ShiftedBronzes and other general OOD datasets (Species, ImageNet-O, iNaturalist, Texture, OpenImage-O). The bar chart is arranged in ascending order of distribution shift from left to right. We report only the average FPR95 across general OOD datasets.

are 4:1:5 (2278:572:2861), respectively. And each fine-grained period and bronze category is split following this ratio.

The uniformity of background information in bronze images may limit the effectiveness of OOD detection methods that rely on background cues for training. To mitigate this issue, we constructed an additional ID training set by replacing the original backgrounds of training images with 2,278 randomly selected images from the ImageNet-1K dataset. This synthetic dataset also enables a more comprehensive analysis of the role of background information in model performance.

Besides the ShiftedBronzes OOD data, we additionally utilized five general OOD datasets from ImageNet-1K, including Species (Basart et al., 2022), ImageNet-O (Hendrycks et al., 2021b), iNaturalist (Huang & Li, 2021), Texture (Kylberg, 2011), and OpenImage-O (Wang et al., 2022).

## 4.2 IMPLEMENTATION DETAILS

**Devices and Code.** All experiments were implemented by PyTorch, and conducted on a server with 4 RTX A40 GPUs and Intel® Xeon® Gold 5220 CPUs (72 cores). The code for all post-hoc OOD detection methods and OpenGAN method are derived from openOOD benchmark (Yang et al., 2022a). And the code for other methods are derived from the corresponding official repositories. The hyperparameters settings for each method are provided in the appendix.

**Evaluation.** For evaluation metrics, we employ FPR95 and AUROC for the OOD detection task, overall accuracy (OA) and independent accuracy for the dating task. AUROC is a metric that computes the area under the receiver operating characteristic curve. A higher value indicates better detection performance. FPR95 is short for FPR@TPR95, which is the false positive rate when the true positive rate is 95%. Smaller FPR95 implies better performance. We evaluate the overall performance of different bronze dating methods using OA and assess their individual performance at various levels of granularity using independent accuracy.

To quantify sensitivity of OOD detection methods to varying degrees of distributional discrepancies, we propose the distribution-aware degree (DAD), a novel metric evaluating whether an OOD detector can distinguish gradual distribution shifts. Given a progressively ordered sequence of $n$ OOD test sets $\{D_i\}_{i=1}^{n}$ with increasing distribution shift magnitudes from the ID data, we define DAD as:

$$\text{DAD} = \sum_{j=1}^{n-1} \sum_{k=j+1}^{n} \mathbb{I}\left(s_j > s_k\right), \tag{1}$$

where $s_i$ denotes the FPR95 score of a method on OOD test sets $D_i$ and $\mathbb{I}\left(\cdot\right)$ is the indicator function. The metric ranges from 0 (worst) to $\frac{n(n-1)}{2}$ (ideal), reflecting the number of pairwise comparisons where the detector shows distribution hierarchy awareness. We evaluate on 5 distribution shift degree with the optimal DAD value being 10. A higher DAD score indicates a stronger ability to perceive distributional shift.

### 4.3 BRONZE DATING EXPERIMENTS

**Setting.** We select 6 representative FGVC methods to evaluate their bronze dating performance on the ShiftedBronzes. Based on the experimental settings in Zhou et al. (2023), we chose three multi-granularity FGVC methods (YourFL (Chang et al., 2021), HRN (Chen et al., 2022), AKG (Zhou et al., 2023)) and three single-granularity FGVC methods (NTS-Net (Yang et al., 2018), SPS (Huang et al., 2021b), P2PNet (Yang et al., 2022b)) for the bronze dating task.

**Results and Analysis.** **(1)** *Domain-specific expert knowledge significantly enhances the model learning for bronze dating.* As can be seen from Table 2, the AKG achieves the best OA performance. Moreover, it also attains the best independent accuracy across two bronze categories, three coarse-granularity eras and five fine-granularity eras. Designed with the dating of bronze Ding, the AKG outperform other general FGVC methods even when applied to the expanded data with Gui. **(2)** *Multi-level feature enhancement does not always yield effective improvements.* Except for the top-performing AKG, the other two multi-granularity FGVC methods underperform compared to SPS and P2PNet. And the performance of HRN on bronze category classification is significantly lower than that of other multi-granularity methods. This indicates that merely obtaining additional information from hierarchical labels is insufficient.

### 4.4 OOD DETECTION EXPERIMENTS

**Setting** We selected sixteen widely studied OOD detection methods, which are divided into two categories (post-hoc, VLM-based), and conducted OOD detection experiments on ShiftedBronzes dataset, as well as five general OOD datasets. The post-hoc methods include DICE (Sun & Li, 2022), EBO (Liu et al., 2020), GradNorm (Huang et al., 2021a), Gram (Sastry & Oore, 2020), KL-Matching (Basart et al., 2022), KNN (Sun et al., 2022), MDS (Lee et al., 2018), MLS (Basart et al., 2022), MSP (Hendrycks & Gimpel, 2017), ODIN (Liang et al., 2018), OpenMax (Bendale & Boult, 2016), ReAct (Sun et al., 2021) and VIM (Wang et al., 2022). We selected the AKG, which demonstrated the best performance in the bronze dating experiments, to serve as the pre-model for post-hoc methods. The VLM-based methods include ID-like (Bai et al., 2024), LoCoOp (Miyai et al., 2024) and CLIPN (Wang et al., 2023). ID-like (Bai et al., 2024) and LoCoOp (Miyai et al., 2024) employ few-shot prompt learning on origin bronze training set. We also conducted experiments to evaluate their performance under varying numbers of training samples and selected the best-performing configurations to compare with other methods. CLIPN (Wang et al., 2023) has been pre-training on the CC3M (Sharma et al., 2018) dataset, followed by zero-shot inference.

**Results and Analysis** **(1)** *Prompt tuning improves the OOD detection performance of VLM-based methods in specialized domain.* As shown in Table 3, ID-like (Bai et al., 2024) consistently achieves the highest performance across most ShiftedBronzes OOD datasets, both in individual evaluations and on average. Similarly, the LoCoOp (Miyai et al., 2024) method, which is also based on VLMs, secures the second-best performance in the majority of cases. However, another VLM-based approach, CLIPN (Wang et al., 2023), demonstrates a notable performance gap compared to ID-like and LoCoOp. These results indicate that, despite the extensive prior knowledge embedded

| Method | Type | Transferred Container FPR95↓ AUROC↑ | Sketch and Rubbing FPR95↓ AUROC↑ | Transferred Bronze FPR95↓ AUROC↑ | Container FPR95↓ AUROC↑ | Average FPR95↓ AUROC↑ |
|---|---|---|---|---|---|---|
| LoCoOp (Miyai et al., 2024) | VLM-based | 30.72/92.25 | 8.45/98.49 | 7.09/98.63 | 10.39/97.02 | 16.41/95.93 |
| ID-like (Bai et al., 2024) | (origin bg) | 22.72/94.43 | 24.55/94.94 | 5.13/98.8 | 0.55/99.79 | 13.24/96.99 |
| LoCoOp (Miyai et al., 2024) | VLM-based | 28.45/91.79 | 15.29/96.95 | 4.5 /98.98 | 6.93 /98.14 | 15.44/96.06 |
| ID-like (Bai et al., 2024) | (outlier bg) | 6.4 /98.01 | 14.89/95.46 | 1.24 /99.59 | 0.09 /99.97 | 5.66 /98.26 |

Table 4: Prompt tuning with background-replaced bronze images resulted in notable performance improvements for ID-like and LoCoOp across most OOD data in the ShiftedBronzes, enhancing both individual and average performance (marked with blue substrate).

in VLMs, their direct application to domain-specific OOD detection remains suboptimal. Effective prompt learning is essential to infuse domain-specific knowledge and enhance OOD detection performance in specialized domains. **(2)***VLM-based methods consistently outperform post-hoc methods in specialized domain OOD detection.* Our comparative analysis of post-hoc and VLM-based methods demonstrates that VLM-based approaches consistently outperform post-hoc methods in domain-specific OOD detection tasks. However, post-hoc methods remain attractive due to their ease of deployment and the lack of a need for model retraining. This underscores the importance of developing more advanced post-hoc methods specifically designed for domain-specific OOD detection, ideally achieving performance on par with VLM-based methods. **(3)** *Methods with higher distribution-aware degree values tend to achieve better OOD performance.* We further analyze the DAD under the FPR95 metric, as shown in Figure 6 (left). A clear trend emerges: methods with higher DAD values tend to achieve better OOD performance. To explore this relationship, we categorize the methods based on their DAD values and visualize their FPR95 scores across different OOD datasets in Figure 6 (right). The results provide insights into the varying degrees of distribution shift awareness exhibited by different methods. **(4)** *The SOTA OOD detection method remains for further enhancing in covariate shift detection.* As illustrated in Figure 6, most methods struggle to detect OOD samples in the sketch and rubbing of ShiftedBronzes, which exhibit covariate shifts. This suggests that current OOD detection techniques are generally insensitive to covariate shifts, even when such shifts result in greater distributional divergence from ID data than shifts in transferred data. This highlight the need for more effective OOD methods that can robustly handle covariate shifts in fine-grained domains. **(5)** *It is necessary to specifically enhance the ability to detect color and background shifts.* Comparing the types of distributional shifts observed in Figure 1, sketch and rubbing images exhibit clear color and background shifts relative to the ID data. This suggests that improving the sensitivity of OOD detection methods to color and background changes could enhance their overall ability to detect covariate shifts. **(6)** *Gram matrix effectively capture the distributional differences across various distribution shift.* As shown in Figure 6 (b), although the Gram (Sastry & Oore, 2020) method performs poorly on the ShiftedBronzes benchmark, it achieves a notably high DAD score of 9. This suggests that the Gram matrix effectively captures distributional differences among various OOD datasets, but struggles to distinguish ID from OOD samples. These findings suggest that, despite its limitations in detecting fine-grained distribution shifts, the Gram matrix's sensitivity to varying degrees of shift can be leveraged to enhance the distributional awareness of post-hoc methods. **(7)** *A clear performance gap is observed between general-purpose and domain-specific OOD detection benchmarks.* ID-like, the best-performing method on Shifted-Bronzes, achieves a remarkably low average FPR95 of 0.23% and a near-perfect average AUROC of 99.92% across 5 general OOD benchmarks. However, its performance on ShiftedBronzes still leaves room for improvement. This highlights that performance on general-purpose datasets alone is insufficient to evaluate OOD robustness in specialized domains, and that more rigorous evaluation on domain-specific OOD data is necessary. Full experimental results on general OOD datasets are provided in the appendix.

## 4.5 BACKGROUND INFLUENCE

**Setting** ID-like (Bai et al., 2024) and LoCoOp (Miyai et al., 2024) leverage the background of training images for OOD prompt learning. While demonstrating effectiveness on general-purpose datasets with diverse background variations, its applicability to domain-specific datasets is constrained by the typically homogeneous nature of background information. To investigate this limitation, we evaluate the performance of ID-like and LoCoOp using composite images with replaced backgrounds as training samples.

**Results and Analysis** *The inherent limitations of VLM-based OOD detection methods in specialized domains can be mitigated by training with background-replaced ID data.* As shown in Table 4, prompt tuning with background-replaced bronze images resulted in notable performance improvements for ID-like (Bai et al., 2024) across all OOD data in the ShiftedBronzes, enhancing both individual and average performance. Similarly, LoCoOp (Miyai et al., 2024) demonstrated improved performance on three OOD datasets, along with an increase in its overall average performance. These findings highlight the limitations of VLM-based OOD detection methods when applied to domain-specific datasets, particularly those reliant on background information. While simple background replacement serves as an effective mitigation strategy, further improvements in domain-specific OOD detection require the incorporation of additional outlier data.

## 4.6 SAMPLE SIZE INFLUENCE

**Setting** We systematically evaluated the ID and OOD performance of ID-like (Bai et al., 2024) and LoCoOp (Miyai et al., 2024) across different training sample sizes. The experiments commenced from one-shot setting and incrementally expanded the training samples size until the full training set dataset was utilized. We also considered two training configurations: one using original-background samples and the other employing replaced-background samples.

**Results and Analysis** *Increasing the training sample size enhances the ID performance of VLM-based methods in specialized domains while may impact OOD performance.* Figure 7 illustrates that while ID-like (Bai et al., 2024) and LoCoOp (Miyai et al., 2024) demonstrate strong OOD detection performance, their ID accuracy remains considerably lower than that of the best-performing FGVC methods. Furthermore, as the number of training samples increases, both methods exhibit improved ID accuracy when trained on original-background data, whereas their OOD detection performance gradually deteriorates. Notably, LoCoOp achieves higher ID accuracy than ID-like but suffers from a more pronounced decline in OOD performance. Conversely, when trained with varying amounts of replaced-background samples, ID-like not only continues to enhance its ID accuracy but also stabilizes its OOD performance near the optimal level. Moreover, ID-like can further boost its OOD performance by incorporating additional outlier data from replaced-background.

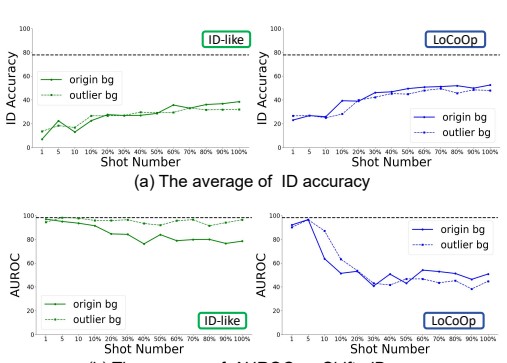

(a) The average of ID accuracy

(b) The average of AUROC on ShiftedBronzes

Figure 7: Comparison of ID and OOD performance of ID-like (Bai et al., 2024) and LoCoOp (Miyai et al., 2024) under varying amounts of training data. The black dashed line in the figure represents the best ID accuracy and OOD AUROC achieved among all compared methods, serving as a reference for evaluating the relative performance of each method.

## 5 CONCLUSION AND LIMITATION

In this work, we introduce ShiftedBronzes, the first benchmark explicitly designed to evaluate OOD detection under fine-grained, multi-level distribution shifts. Our study reveals critical limitations in current OOD methods: post-hoc detectors often fail to respond to subtle distributional changes, and VLM-based methods are prone to background bias under prompt tuning. To facilitate deeper understanding, we propose a distribution-aware sensitivity metric, which quantifies a method's responsiveness to varying degrees of shift and correlates with detection effectiveness. Our benchmark and analysis offer both diagnostic insights and future directions for building OOD detection systems that are robust to the nuanced shifts commonly found in real-world applications.

While our study provides new insights into fine-grained OOD detection under progressive distribution shifts, several limitations remain. First, while ShiftedBronzes adopts bronze dating as a proxy for multi-level fine-grained OOD evaluation, the framework and insights are expected to generalize with further validation. Second, the proposed distribution-aware sensitivity metric offers useful diagnostics, and future work may explore leveraging it to improve OOD detection performance.

## REPRODUCIBILITY STATEMENT

Dataset creation and processing steps are described in Section 3 and Appendix A.1. Implementation details are described in Sections 4.2 and Appendix A.2, including model architecture, training hyperparameters, and evaluation protocols. The code and dataset will be made publicly available in a future release.

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

# A APPENDIX

## A.1 ANNOTATION DETAILS

**Bronze, Sketch and Rubbing Data.** Building on the annotation workflow proposed by Zhou et al. (2023) for bronze Ding, we employed the LabelImg tool to annotate the characteristics of Gui using bounding boxes, as shown in Figure 8. These annotations provide archaeologists with a systematic tool to explore correlations between bronze dating and decorative patterns. Additionally, they support the development of advanced algorithms for bronze dating by offering a robust dataset for pattern recognition and analysis. The collection and labelling of data were carried out by an archaeologist and eight archaeology assistants, who took approximately six months to complete. Because many dating results are controversial, we re-argue the era of each artifact through discussions with three bronze experts.

We summarize the distribution of shape and attribute annotations in Figure 2 (a) and (b) of the main paper. A complete list of all shape and characteristic categories, along with their corresponding frequencies, is provided in Table 8 and Table 9. The dataset contains a wide variety of shapes and characteristics, with a long-tailed distribution. Bowl-shaped, jar-shaped, and round ding types are the most common among the shape annotations, while several specialized shapes (e.g., "Hoof-footed Square Ding", "Conical-footed Square Ding") are rare. Pillar foot, Upright ear, and Ring ear types are the most common among the characteristic annotations, while several specialized shapes (e.g., "Tray", "Bird claw-shaped hook pendant") are rare.

**Container Data.** We summarize the distribution of containers in Figure 2 (c) of the main paper. A complete list of all container categories, along with their corresponding frequencies, is provided in Table 10. The dataset exhibits a heavy-tailed distribution, where common categories such as "bowl", "vacuum flask", and "liquor glass" dominate, while many fine-grained types (e.g., "altar wine", "jug wine", "cupule") appear infrequently.

## A.2 IMPLEMENTATION DETAILS

### A.2.1 BRONZE DATING METHODS

Table 5 summarizes the hyperparameter configurations used for all compared FGVC methods in bronze dating experiments. These include key parameters such as batch size, optimizer type, initial learning rate, number of training epochs and loss functions, with settings sourced from the official implementations. To ensure fair comparisons, we followed the recommended configurations provided by the authors of each method. This alignment minimizes variability introduced by hyperparameter differences, allowing for a more reliable evaluation of method performance.

### A.2.2 OOD DETECTION METHODS

**Post-hoc.** We performed OOD detection experiments using the post-hoc methods code provided by OpenOOD (Yang et al., 2022a) (pytorch), with parameter settings based on the default configuration tested in OpenOOD.

**VLM-based.** ID-like (Bai et al., 2024) (pytorch) and LoCoOp (Miyai et al., 2024) (pytorch) employ the official code, with parameter settings based on the default configuration reported in corresponding paper. CLIPN (Wang et al., 2023) (pytorch) loads the pretrained weights provided by the official repository to perform zero-shot inference.

## A.3 ADDITIONAL EXPERIMENTS

### A.3.1 BRONZE DATING EXPERIMENTS

In our main paper, we reported the chronological classification accuracy achieved by combining the Ding and Gui datasets. To further investigate their individual performance, we calculated the dating accuracy separately for each dataset, as shown in Tables 6. Notably, owing to social changes in Warring States period, Gui from the Warring States period were extremely scarce. As shown in

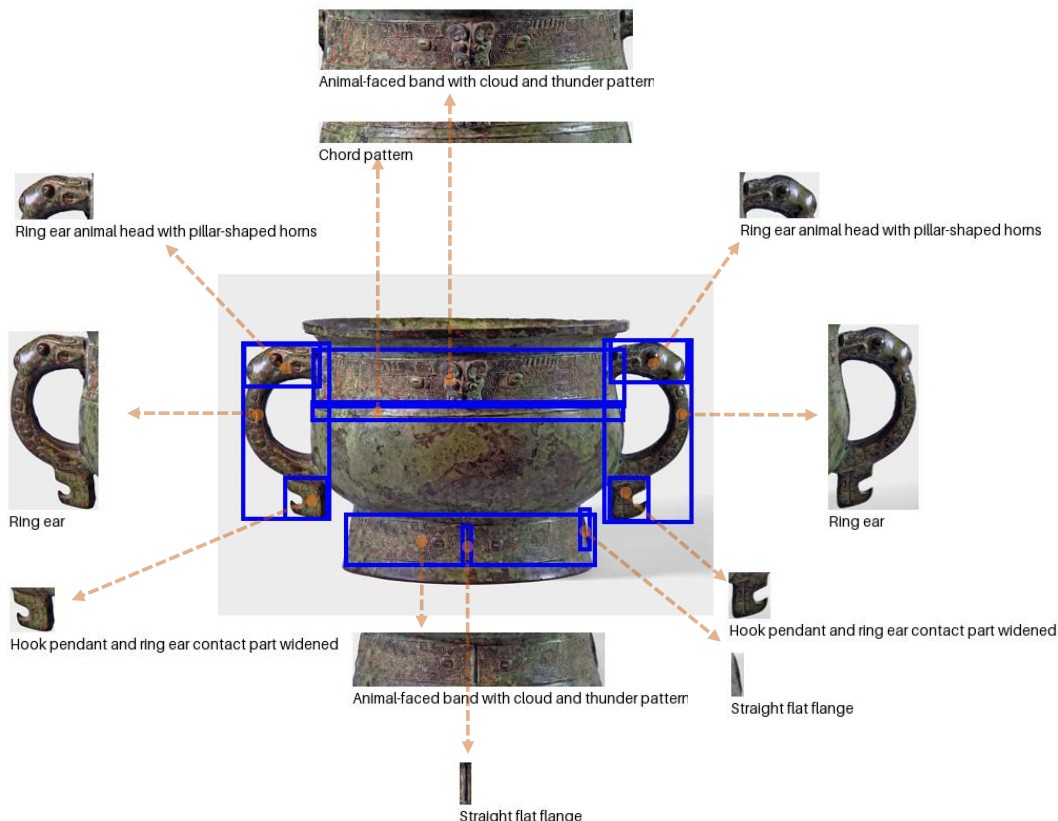

Figure 8: Examples of bounding box annotations for characteristics on bronze Gui images in the ShiftedBronzes.

| | Backbone | Batch Size | Optimizer | Learning Rate | Epochs | Loss Function |
|---|---|---|---|---|---|---|
| NTS-Net (Yang et al., 2018) (pytorch) | ResNet-50 | 16 | SGD | 1e-3 | 500 | Cross Entropy Loss |
| SPS (Huang et al., 2021b) (pytorch) | | 16 | SGD | 1e-3 | 160 | Cross Entropy Loss |
| P2PNet (Yang et al., 2022b) (pytorch) | | 32 | SGD | 2e-3 | 300 | Cross Entropy Loss |
| YourFL (Chang et al., 2021) (pytorch) | | 28 | SGD | 1e-1 | 150 | Cross Entropy Loss, Neg Log-Likelihood Loss |
| HRN (Chen et al., 2022) (pytorch) | | 8 | SGD | 1e-4 | 200 | Cross Entropy Loss, Tree Loss |
| AKG (Zhou et al., 2023) (pytorch) | | 64 | Adam | 1e-4 | 128 | Cross Entropy Loss, Graph Loss |

Table 5: Hyperparameter settings for all compared FGVC methods.

Table 1 of the main text, the bronze Gui includes only five images from the mid Warring States period and a single image from the late Warring States period. Due to this severe class imbalance, we omitted dating for Gui from the Warring States period, with results denoted by a dash in the Table 6 (b).

Table 6 (a) summarizes the dating accuracy for the Ding data across different granularities. AKG (Zhou et al., 2023) consistently achieves the highest overall accuracy (OA) across three granularity levels and outperforms other methods for the bronze category and four coarse-granularity eras. For the 11 fine-granularity eras, AKG and P2PNet (Yang et al., 2022b) each excel in six eras, with one era yielding identical performance. These findings suggest that AKG's robust feature extraction and classification capabilities make it an effective pre-classifier for enhancing post-hoc methods in bronze OOD detection tasks.

Table 6 (b) reports the dating accuracy for Gui data across different granularities. AKG achieves the highest OA across all three granularity levels, demonstrating its robustness in handling both coarse- and fine-grained bronze dating tasks. However, for the four coarse-grained eras and 11 fine-grained

eras, specific methods excel in certain periods, reflecting the varying strengths of these approaches. Notably, YourFL (Chang et al., 2021) achieves an accuracy of 0 for the Early Shang period, in stark contrast to other methods that perform significantly better during this era. This discrepancy can be attributed to two primary factors. First, the severe data imbalance for under-represented eras, such as the Early Shang, disproportionately impacts YourFL. Second, YourFL's hierarchical interaction structure appears less effective at utilizing coarse-grained features to improve fine-grained feature learning compared to the adaptive mechanisms employed by HRN (Chen et al., 2022) and AKG. These findings underscore the importance of balanced data distribution and feature integration strategies in enhancing model performance across granularity levels.

| | | | Ding | | | | | | | | | | |
|---|---|---|---|---|---|---|---|---|---|---|---|---|---|
| | Method | OA | Shang | | Western Zhou | | | Spring and Autumn | | | Warring States | | |
| | | | Early | Late | Early | Mid | Late | Early | Mid | Late | Early | Mid | Late |
| Single-Granularity | NTS-Net (Yang et al., 2018) | 74.82 | 80.85 | 79.07 | 78.55 | 80.49 | 82.58 | 72.48 | 57.89 | 71.3 | 30.23 | 35.9 | 71.83 |
| | SPS (Huang et al., 2021b) | 77.82 | 80.85 | 82.24 | 78.3 | **87.32** | 81.06 | 80.54 | 60.0 | 67.59 | 51.16 | 35.9 | 83.1 |
| | P2PNet (Yang et al., 2022b) | 77.08 | **87.23** | 70.19 | **85.54** | 84.39 | 81.82 | **85.91** | 55.79 | **75.0** | 48.84 | **41.03** | **88.73** |
| Multi-Granularity | YourFL (Chang et al., 2021) | 99.6 | 99.6 | | | | | 99.6 | | | | | |
| | | 81.79 | 78.27 | | 83.88 | | | 84.38 | | | 77.78 | | |
| | | 72.6 | 74.47 | 78.01 | 73.32 | 81.95 | 76.52 | 74.5 | 56.84 | 63.89 | 27.91 | **41.03** | 71.83 |
| | HRN (Chen et al., 2022) | 25.24 | | | | | | 25.24 | | | | | |
| | | 86.5 | 82.88 | | 88.48 | | | 88.92 | | | 83.66 | | |
| | | 75.78 | 78.72 | 79.7 | 79.3 | 80.0 | 79.55 | 80.54 | 57.89 | 70.37 | 39.53 | 23.08 | 81.69 |
| | AKG (Zhou et al., 2023) | **99.66** | | | | | | **99.66** | | | | | |
| | | **88.14** | **85.19** | | **90.51** | | | **89.2** | | | **84.31** | | |
| | | **79.75** | **87.23** | 83.3 | 82.04 | **87.32** | 83.33 | 79.87 | **65.26** | 70.37 | **53.49** | 33.33 | 84.51 |

(a) Dating results on bronze Ding

| | | | Gui | | | | | | | | | | |
|---|---|---|---|---|---|---|---|---|---|---|---|---|---|
| | Method | OA | Shang | | Western Zhou | | | Spring and Autumn | | | Warring States | | |
| | | | Early | Late | Early | Mid | Late | Early | Mid | Late | Early | Mid | Late |
| Single-Granularity | NTS-Net (Yang et al., 2018) | 70.55 | **100.0** | 58.99 | 80.05 | 70.18 | 77.27 | 37.68 | 25.0 | 71.43 | - | - | - |
| | SPS (Huang et al., 2021b) | 72.38 | **100.0** | 73.03 | 80.33 | 67.64 | 72.73 | 47.83 | **37.5** | 78.57 | - | - | - |
| | P2PNet (Yang et al., 2022b) | 74.31 | **100.0** | 69.66 | **83.88** | 68.73 | 69.89 | **72.46** | 25.0 | 78.57 | - | - | - |
| Multi-Granularity | YourFL (Chang et al., 2021) | 99.45 | | | | | | 99.45 | | | | | |
| | | 83.3 | 56.59 | | 92.04 | | | 58.24 | | | - | | |
| | | 66.51 | 0.0 | 60.11 | 79.78 | 62.18 | 62.5 | 55.07 | 0.0 | 50.0 | - | - | - |
| | HRN (Chen et al., 2022) | 62.75 | | | | | | 62.75 | | | | | |
| | | 85.13 | 64.84 | | **92.66** | | | 58.24 | | | - | | |
| | | 70.73 | 75.0 | 64.04 | 81.15 | **71.64** | 63.07 | 53.62 | 12.5 | 78.57 | - | - | - |
| | AKG (Zhou et al., 2023) | **99.82** | | | | | | **99.82** | | | | | |
| | | 87.98 | **75.82** | | 92.41 | | | **72.53** | | | - | | |
| | | 74.86 | **100.0** | 75.28 | 82.79 | 66.55 | **76.7** | 63.77 | **37.5** | 71.43 | - | - | - |

(b) Dating results on bronze Gui

Table 6: The comparison of six FGVC methods on the (a) bronze Ding and (b) bronze Gui. In addition to the overall accuracy on the test set, we have also tested each accuracy of different independent era and bronze category. Bold indicates the best results. The dash in (b) indicates that accuracy for this specific age is not calculated.

### A.3.2 OOD Detection Experiments

**Degradation Analysis**    To investigate the reasons behind the failure of OOD detection methods on the ShiftedBronzes, we conducted a detailed analysis of the least effective methods. Among the 16 competing methods, KL-Matching (Basart et al., 2022) and DICE (Sun & Li, 2022) exhibited the poorest performance. We derived two primary conclusions. **(1)** *Class imbalance adversely affect methods that rely on category-level statistics.* KL-Matching calculates OOD scores using a posterior distribution template based on the maximum posteriori probability of each class in the validation set. However, class imbalance in the bronze dataset may cause the posterior template to fail to capture certain class characteristics, impairing OOD sample detection during inference. **(2)** *Methods tailored for small-scale datasets struggle to cope with specialized domain data.* OOD detection methods on small-scale datasets struggle with large semantic spaces in open-world scenarios. As data scale and specificity increase, distribution shifts and noise patterns become more complex. DICE, designed for small-scale datasets, does not account for large-scale OOD datasets.

**Score Density Distribution**    Figures 9, 10, 11, 12, 13, 14, 15, 16 present the ID and OOD score density distributions for all 16 methods across various OOD datasets, ranked by their distribution-aware degree (DAD). When comparing different methods, LoCoOp (Miyai et al., 2024), VIM (Wang et al., 2022), MDS (Lee et al., 2018), and ID-like (Bai et al., 2024) exhibit a more pronounced separation between the score distributions of ID and OOD samples. This separation enables these

| Method | Type | Species FPR95↓ AUROC↑ | ImageNet-O FPR95↓ AUROC↑ | iNaturalist FPR95↓ AUROC↑ | Texture FPR95↓ AUROC↑ | OpenImage-O FPR95↓ AUROC↑ | Average FPR95↓ / AUROC↑ |
|---|---|---|---|---|---|---|---|
| DICE (Sun & Li, 2022) | Post-hoc | 84.24/57.12 | 90.93/44.59 | 93.41/31.69 | 92.60/45.89 | 93.15/41.30 | 90.07 / 44.52 |
| EBO (Liu et al., 2020) | | 45.40/88.15 | 59.26/82.93 | 56.91/74.61 | 86.27/69.51 | 60.61/79.88 | 61.69 / 79.82 |
| GradNorm (Huang et al., 2021a) | | 38.78/90.71 | 52.93/84.63 | 52.86/77.49 | 85.72/67.14 | 57.23/81.31 | 57.10 / 80.26 |
| Gram (Sastry & Oore, 2020) | | 80.13/77.78 | 87.14/68.33 | 74.95/65.27 | 83.76/69.40 | 85.69/64.68 | 82.33 / 69.09 |
| KL-Matching (Basart et al., 2022) | | 96.33/65.96 | 96.30/60.98 | 97.75/41.62 | 96.66/54.15 | 96.72/55.56 | 96.75 / 55.25 |
| KNN (Sun et al., 2022) | | 17.81/95.98 | 22.09/94.61 | 18.65/94.10 | 28.04/93.03 | 22.35/94.56 | 21.79 / 94.46 |
| MDS (Lee et al., 2018) | | 10.03/96.95 | 7.11/98.11 | 1.64/99.48 | 1.83/99.61 | 6.88/98.34 | 5.50 / 98.50 |
| MLS (Basart et al., 2022) | | 45.66/87.82 | 59.42/82.64 | 56.91/74.43 | 86.27/69.61 | 60.68/79.67 | 61.79 / 78.83 |
| MSP (Hendrycks & Gimpel, 2017) | | 54.57/80.32 | 63.76/76.17 | 64.02/69.84 | 75.56/71.23 | 63.73/74.39 | 64.33 / 74.39 |
| ODIN (Liang et al., 2018) | | 80.48/82.88 | 94.50/74.69 | 79.61/81.23 | 96.40/71.88 | 87.91/79.27 | 87.38 / 78.39 |
| OpenMax (Bendale & Boult, 2016) | | 55.50/79.80 | 64.86/76.79 | 65.24/69.78 | 74.60/71.81 | 64.69/74.91 | 64.98 / 74.62 |
| ReAct (Sun et al., 2021) | | 47.52/87.17 | 60.55/81.34 | 58.97/72.51 | 87.78/68.41 | 61.32/78.42 | 63.63 / 77.57 |
| VIM (Wang et al., 2022) | | 4.47/98.97 | 4.47/98.96 | 1.41/99.72 | 1.29/99.59 | 4.50/99.03 | 3.23/99.25 |
| LoCoOp (Miyai et al., 2024) | VLM-based | 19.37/94.36 | 5.63/98.48 | 1.85/99.56 | 4.17/98.96 | 10.65/97.32 | 8.33 / 97.74 |
| ID-like (Bai et al., 2024) | | **0.07/99.93** | **0.03/99.95** | **0.0/99.99** | **0.0/99.98** | **0.23/99.92** | **0.07 / 99.95** |
| CLIPN (Wang et al., 2023) | | 7.25/98.11 | 7.71/98.12 | 21.82/94.26 | 8.46/97.59 | 13.53/96.25 | 11.75 / 96.87 |

Table 7: The comparison of sixteen OOD detection methods on the 5 general OOD datasets. We categorized the comparative methods into two types based on their mechanisms and reported their FPR95 and AUPOC. Bold indicates the best performance, while underlined denotes the second-best performance.

methods to more effectively detect OOD inputs. In contrast, other methods show substantial overlap between ID and OOD scores, suggesting potential challenges in accurately distinguishing the two. The score density of VIM (Wang et al., 2022), MDS (Lee et al., 2018), KNN (Sun et al., 2022), and ID-like (Bai et al., 2024) reveal that the ID and ShiftedBronze OOD distributions are less separated compared to those from general OOD datasets, suggesting that existing OOD detection approaches remain challenged by subtle domain shifts.

**ShiftedBronze vs. General OOD Dataset** Table 7 reports the performance of all compared methods on five general-domain OOD datasets. Among them, ID-like (Bai et al., 2024) achieves the best results, with an average FRP@95 of 0.07 and an average AUROC of 99.95. Compared to its performance on the ShiftedBronze benchmark, this represents an improvement of 13.17 in FRP@95 and 2.96 in AUROC. These results suggest that even methods that achieve state-of-the-art performance on general-domain OOD benchmarks still face challenges when applied to domain-specific OOD scenarios such as ShiftedBronze.

LARGE LANGUAGE MODEL (LLM) USAGE

Parts of the manuscript were polished for grammar and style using LLM under the authors' direction. The authors verified and edited all generated text, and the model was not involved in generating research ideas, experimental design, or results.

| Characteristic | Count | Characteristic | Count |
|---|---|---|---|
| Pillar foot | 6188 | Ring ear animal head with angular shape | 146 |
| Upright ear | 5622 | Elephant trunk-shaped flat foot | 145 |
| Ring ear | 3757 | Upright ear outward | 133 |
| Hoof foot | 3279 | Kui dragon pattern curled | 133 |
| Straight flat flange | 2431 | Animal-faced band with others or unclear | 132 |
| Chord pattern | 1800 | Curved ruler knot | 130 |
| Circular vortex pattern | 1791 | Large bird pattern with head up | 127 |
| Animal head on foot | 1480 | Animal-faced pattern with others | 126 |
| Bird head dragon body pattern | 1237 | Ring ear animal head with vertical animal horns | 126 |
| Animal-faced pattern as a single entity | 1112 | Dragon pattern with single head and double body | 124 |
| Ring ear animal head with pillar-shaped horns | 1055 | Slanted angle cloud pattern | 119 |
| Attached ear | 994 | Animal-faced pattern with tail upcurl | 117 |
| Ordinary cloud and thunder pattern | 933 | Bird-shaped flat foot | 113 |
| Tile pattern | 924 | Eye cloud pattern | 111 |
| L-shaped flat foot | 818 | Animal-faced pattern decomposed | 108 |
| Kui dragon pattern with straight body | 805 | Ring ear animal head with retro drum-like animal head | 108 |
| Hook pendant and ring ear contact part widened | 763 | Continuous cicada pattern | 106 |
| Cover ring-shaped handle | 761 | Slanted angle eye pattern | 100 |
| Coiled dragon pattern | 734 | Kui dragon pattern with curled nose | 97 |
| Kui dragon pattern evolved Qiequ pattern | 709 | Bird head dragon body pattern looking back | 92 |
| Widened hook pendant closed to form a polygonal hook pendant | 685 | Pillar foot thick on top and thin at bottom | 91 |
| Cicada pattern foot | 676 | S-shaped stealing patterns | 88 |
| Cicada pattern | 674 | Rope pattern | 75 |
| Attached ear curved | 647 | Interconnected thunder pattern | 75 |
| Double ring pattern | 618 | Ring ear animal head with screen ear animal head | 72 |
| Animal-faced band with cloud and thunder patterns | 587 | Ring ear bird-shaped ear | 66 |
| F-shaped flange | 587 | Reverse hook pendant | 66 |
| Closed elephant trunk-shaped pendant | 577 | Large bird pattern looking back | 57 |
| Three-ring knob | 567 | Elephant trunk-shaped hook pendant | 51 |
| Square pendant with bird shape | 564 | Animal-faced band with divided tail | 48 |
| Cover | 541 | Ring ear S-shaped single animal ear style | 43 |
| Animal-faced pattern with tail downcurl | 527 | Small hook pendant | 40 |
| Ring ear with bird wing pattern in the middle | 509 | Other decorations | 39 |
| Hoof foot short and thick | 456 | Upright bird pattern | 39 |
| Ring ear animal head single animal ear B-type forked horn with tongue out | 446 | Square seat with a gap | 38 |
| Four-petaled eye pattern | 424 | Rhombic lattice pattern | 36 |
| Hoof foot slender | 345 | Relief bird decoration on ring ear | 35 |
| Animal head holding ring ear | 340 | Animal leg-shaped foot | 34 |
| Ring ear animal head rabbit | 339 | Animal-faced pattern with fire pattern | 33 |
| Square pendant other | 320 | Elephant trunk-shaped pendant touching the ground | 33 |
| Glancing dragon pattern with other slanted body and double head | 313 | Cover lotus-shaped handle | 30 |
| Scale pattern | 311 | Square seat with perforations | 28 |
| Panim pattern | 308 | Ring ear dragon shape | 24 |
| U-shaped Qiequ pattern | 306 | Group dragon pattern | 22 |
| Other flanges | 305 | Dragon-shaped ear handle | 21 |
| G-shaped Qiequ pattern | 298 | Elephant head nose foot | 17 |
| Small bird pattern | 296 | Ring ear animal head with small elephant head | 17 |
| Ring ear animal head single animal ear B-type spiral horn | 290 | Ring ear elephant head | 17 |
| Nipple-like thunder pattern | 286 | Elephant trunk-shaped pendant | 16 |
| Kui dragon-shaped flat foot | 284 | Attached ear S-shaped | 15 |
| Straight ridge pattern | 283 | Scattered panim pattern | 14 |
| Triangular pattern | 280 | Decomposed animal-faced pattern with simplified variations | 12 |
| Nipple pattern | 269 | Dragon head bird body | 11 |
| Separated decomposed animal-faced Qiequ pattern | 267 | Elephant head-shaped pendant touching the ground | 10 |
| Ring ear animal head with goat horns | 264 | Tiger-shaped foot | 10 |
| Circular pattern | 260 | Banana leaf pattern | 10 |
| Ring ear animal head single animal ear A | 259 | Ring ear first phase | 9 |
| Animal-faced band with simplified variations | 257 | Long nose-shaped foot | 9 |
| Snake pattern | 257 | Ring ear animal head single animal ear B-type upright pillar-shaped horn | 9 |
| Straight flat foot | 255 | Elephant pattern | 9 |
| Long-tailed bird pattern | 253 | Glancing dragon pattern separated | 8 |
| Conical foot | 227 | Interlaced dragon pattern | 8 |
| Kui dragon pattern with low head and curled tail | 225 | Small bird pattern looking back | 7 |
| Square seat ordinary | 223 | Pendant position with animal leg-shaped foot | 7 |
| Ring ear animal head with animal horns standing separately | 207 | Ring ear flat body dragon shape | 7 |
| Separated C-shaped tail long bird pattern | 205 | Tiger pattern | 7 |
| Animal-faced band with double body trunk | 202 | Fish pattern | 6 |
| Panhu pattern | 198 | Ring ear animal head single animal ear B-type upright pointed ear | 5 |
| Kui dragon pattern with curved body and arched back | 197 | Door and window type stove | 5 |
| Mountain pattern | 180 | Ring ear animal head with cow | 3 |
| Hoof foot flared | 154 | Pendant position with pillar foot | 3 |
| Glancing dragon pattern with folded body | 151 | Ordinary stove | 3 |
| Separated S-shaped tail long bird pattern | 149 | Tray | 2 |
| Other Qiequ pattern | 148 | Bird claw-shaped hook pendant | 1 |

Table 8: Mapping between characteristic category names and their corresponding instance counts in the bronze dataset.

| Shape | Count | Shape | Count |
|---|---|---|---|
| Bowl-shaped | 1410 | Dou-shaped | 53 |
| Jar-shaped | 783 | Waisted Flat-bottomed Ding | 52 |
| Round Ding | 671 | Yue-style Ding B | 28 |
| Hemispherical-bellied Round Ding | 550 | Round Ding with Conical Feet | 27 |
| Column-footed Square Ding | 493 | Narrow-mouthed Ding | 27 |
| Pendent-bellied Round Ding | 413 | Conical-footed Round Ding | 26 |
| Li Ding | 350 | Flat-footed Square Ding | 25 |
| Yu-shaped | 260 | Late Animal-headed Hoof-footed Round Ding | 24 |
| Low and Flat Globular-bellied Ding | 154 | Irregular-shaped Ding | 21 |
| High-Hoof-footed Round Ding | 122 | Yue-style Ding A | 20 |
| Flat-footed Round Ding | 119 | Yi Ding | 19 |
| Hemispherical or Super-hemispherical-bellied Round Ding | 116 | Waisted Round Ding | 8 |
| Early Animal-headed Hoof-footed Round Ding | 107 | Shallow Drum-bellied Ding | 7 |
| Super-hemispherical or Hemispherical-bellied Ding | 97 | Square Gui | 3 |
| Late Animal-headed Hoof-footed Ding | 73 | Special | 3 |
| Pendent-bellied Square Ding | 57 | Hoof-footed Square Ding | 1 |
| Jar-shaped Ding | 54 | Conical-footed Square Ding | 1 |

Table 9: Mapping between shape category names and their corresponding instance counts in the bronze dataset.

| Container | Count | Container | Count |
|---|---|---|---|
| bowl | 2389 | white_wine | 678 |
| vacuum_flask, vacuum_bottle | 1547 | ginger_beer | 666 |
| liqueur_glass | 1305 | varietal, varietal_wine | 663 |
| shot_glass, jigger, pony | 1215 | near_beer | 644 |
| beer_mug, stein | 1202 | sake, saki, rice_beer | 633 |
| cup | 1186 | Burgundy, Burgundy_wine | 623 |
| salad_bowl | 1183 | fishbowl, fish_bowl, goldfish_bowl | 614 |
| punch_bowl | 1173 | Bordeaux, Bordeaux_wine | 611 |
| flute, flute_glass, champagne_flute | 1143 | porter, porter's_beer | 592 |
| thermos, thermos_bottle, thermos_flask | 1122 | Munich_beer, Munchener | 586 |
| catsup_bottle, ketchup_bottle | 1087 | dice_cup, dice_box | 561 |
| ink_bottle, inkpot | 1047 | mulled_wine | 515 |
| bottle | 1046 | parfait_glass | 511 |
| sugar_bowl | 1037 | dessert_wine | 479 |
| Dixie_cup, paper_cup | 1031 | finger_bowl | 467 |
| snifter, brandy_snifter, brandy_glass | 1023 | May_wine | 449 |
| sparkling_wine | 1009 | table_wine | 444 |
| cereal_bowl | 991 | mustache_cup, moustache_cup | 440 |
| whiskey_bottle | 985 | slop_basin, slop_bowl | 419 |
| smelling_bottle | 945 | straw_wine | 399 |
| water_glass | 941 | Rhone_wine | 395 |
| lager, lager_beer | 940 | birch_beer | 333 |
| chamberpot, potty, thunder_mug | 935 | champagne_cup | 316 |
| glass, drinking_glass | 883 | California_wine | 313 |
| fortified_wine | 865 | spruce_beer | 310 |
| pill_bottle | 857 | claret_cup | 304 |
| bock, bock_beer | 839 | grace_cup | 297 |
| root_beer | 820 | Rhine_wine, Rhenish, hock | 242 |
| Weissbier, white_beer, wheat_beer | 816 | Canary_wine | 219 |
| draft_beer, draught_beer | 810 | generic, generic_wine | 167 |
| loving_cup | 802 | specimen_bottle | 106 |
| light_beer | 769 | altar_wine, sacramental_wine | 96 |
| hot-water_bottle, hot-water_bag | 754 | magnetic_bottle | 35 |
| port, port_wine | 733 | jug_wine | 25 |
| highball_glass | 709 | cupule, acorn_cup | 25 |
| blush_wine, pink_wine, rose, rose_wine | 707 | | |

Table 10: Mapping between container category names and their corresponding instance counts in the container dataset.

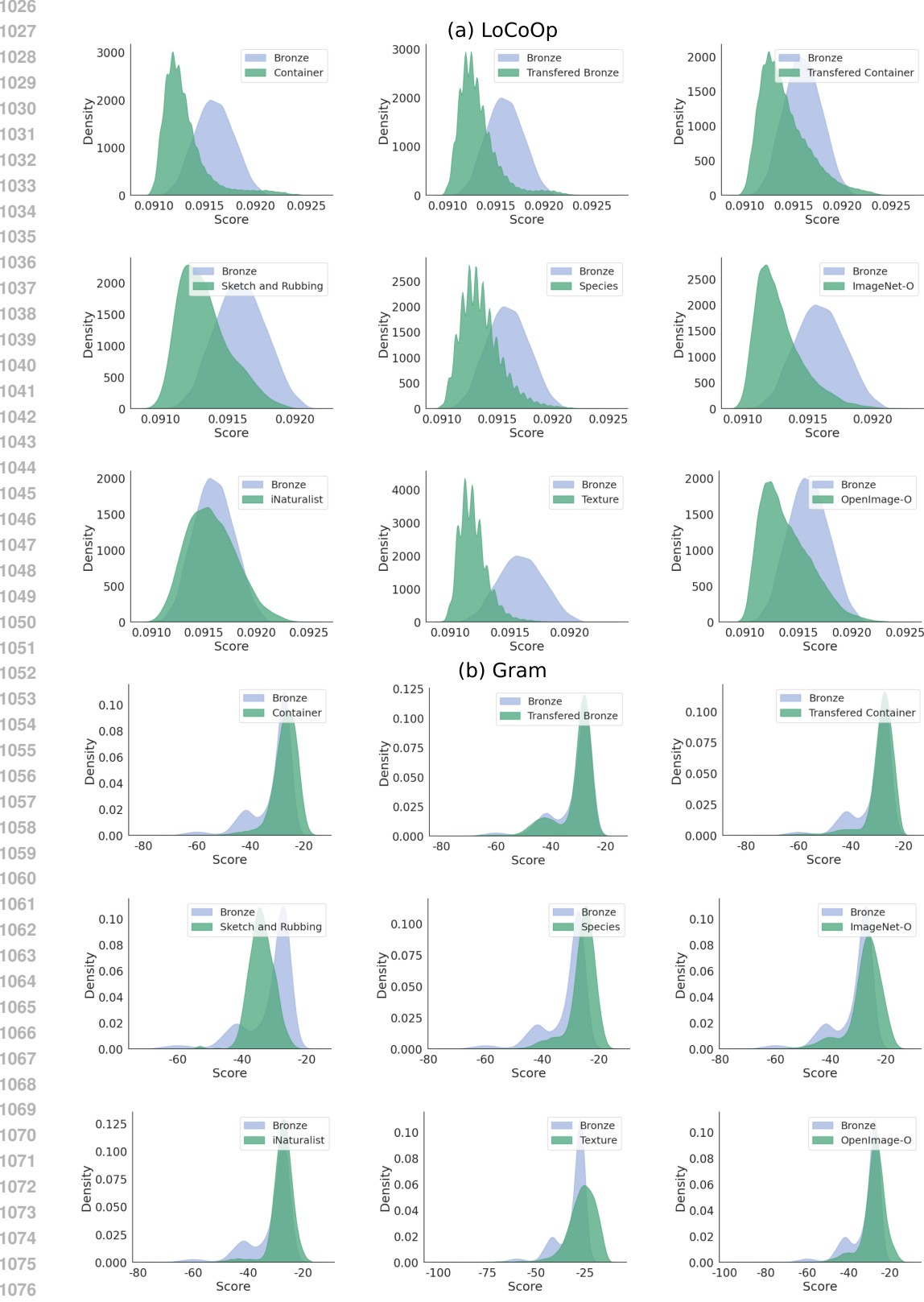

Figure 9: ID and OOD score density distribution of LoCoOp (Miyai et al., 2024) and Gram (Sastry & Oore, 2020) on the ShiftedBronzes and 5 general OOD datasets.

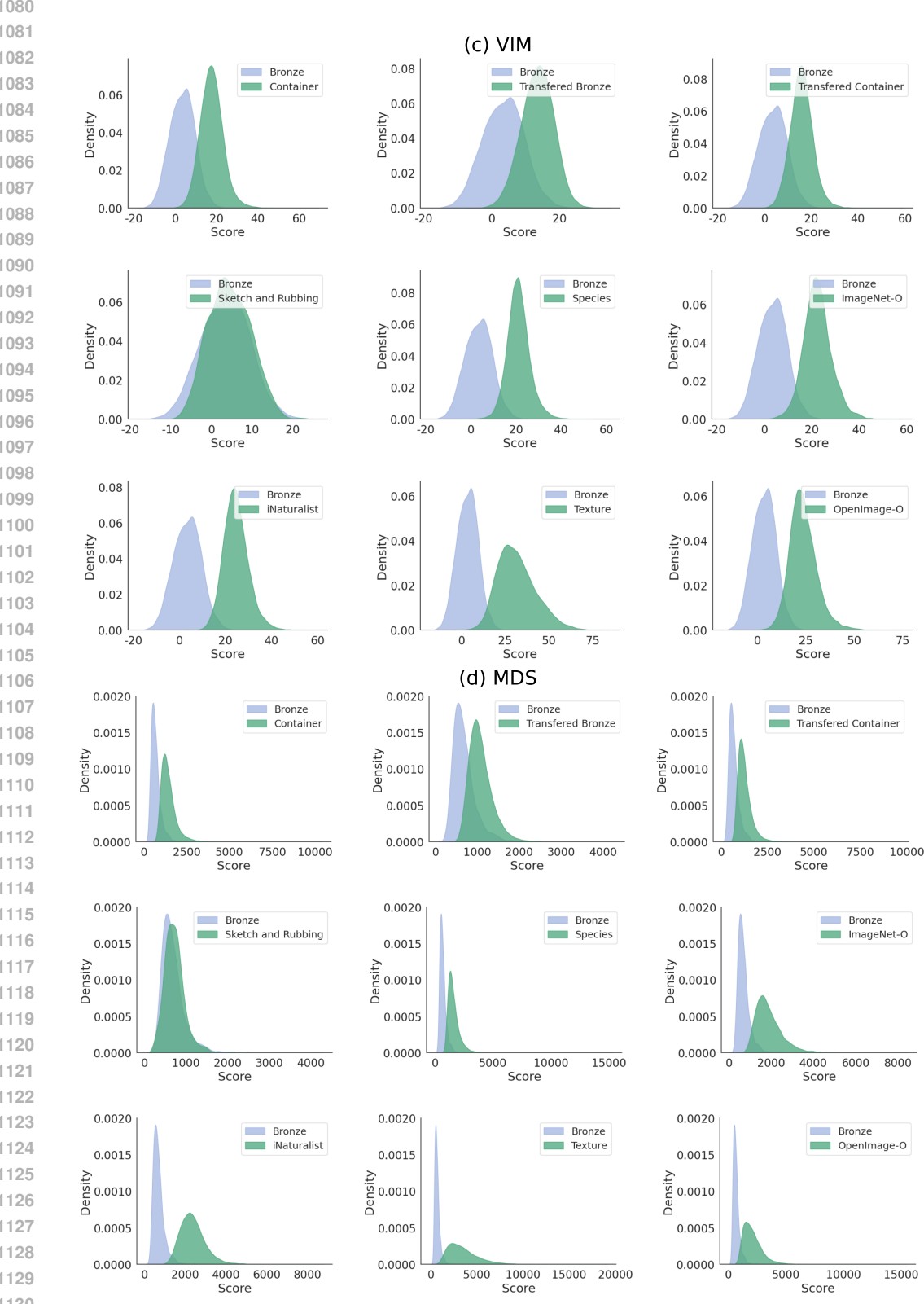

Figure 10: ID and OOD score density distribution of VIM (Wang et al., 2022) and MDS (Lee et al., 2018) on the ShiftedBronzes and 5 general OOD datasets.

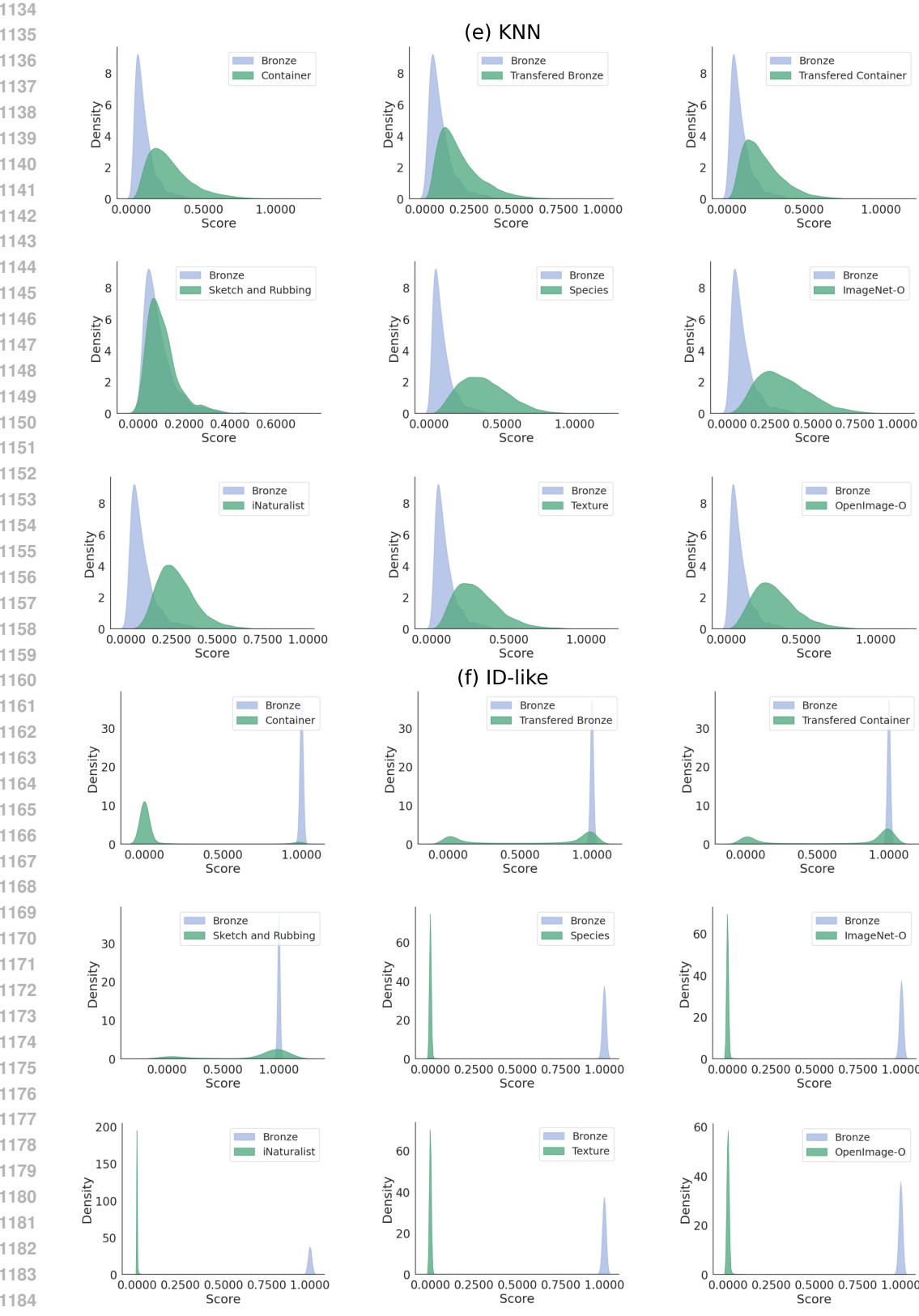

Figure 11: ID and OOD score density distribution of KNN (Sun et al., 2022) and ID-like (Bai et al., 2024) on the ShiftedBronzes and 5 general OOD datasets.

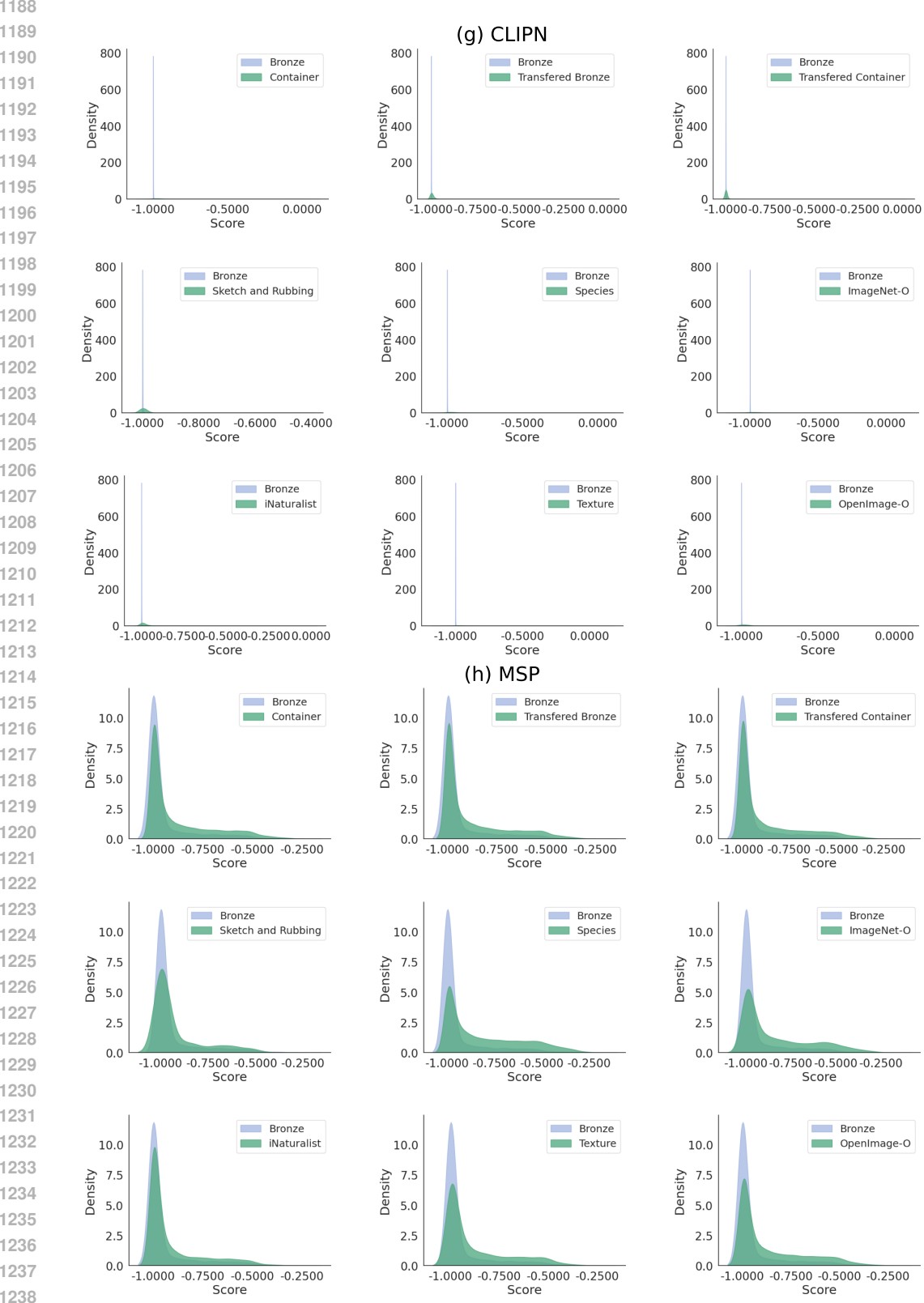

Figure 12: ID and OOD score density distribution of CLIPN (Wang et al., 2023) and MSP (Hendrycks & Gimpel, 2017) on the ShiftedBronzes and 5 general OOD datasets.

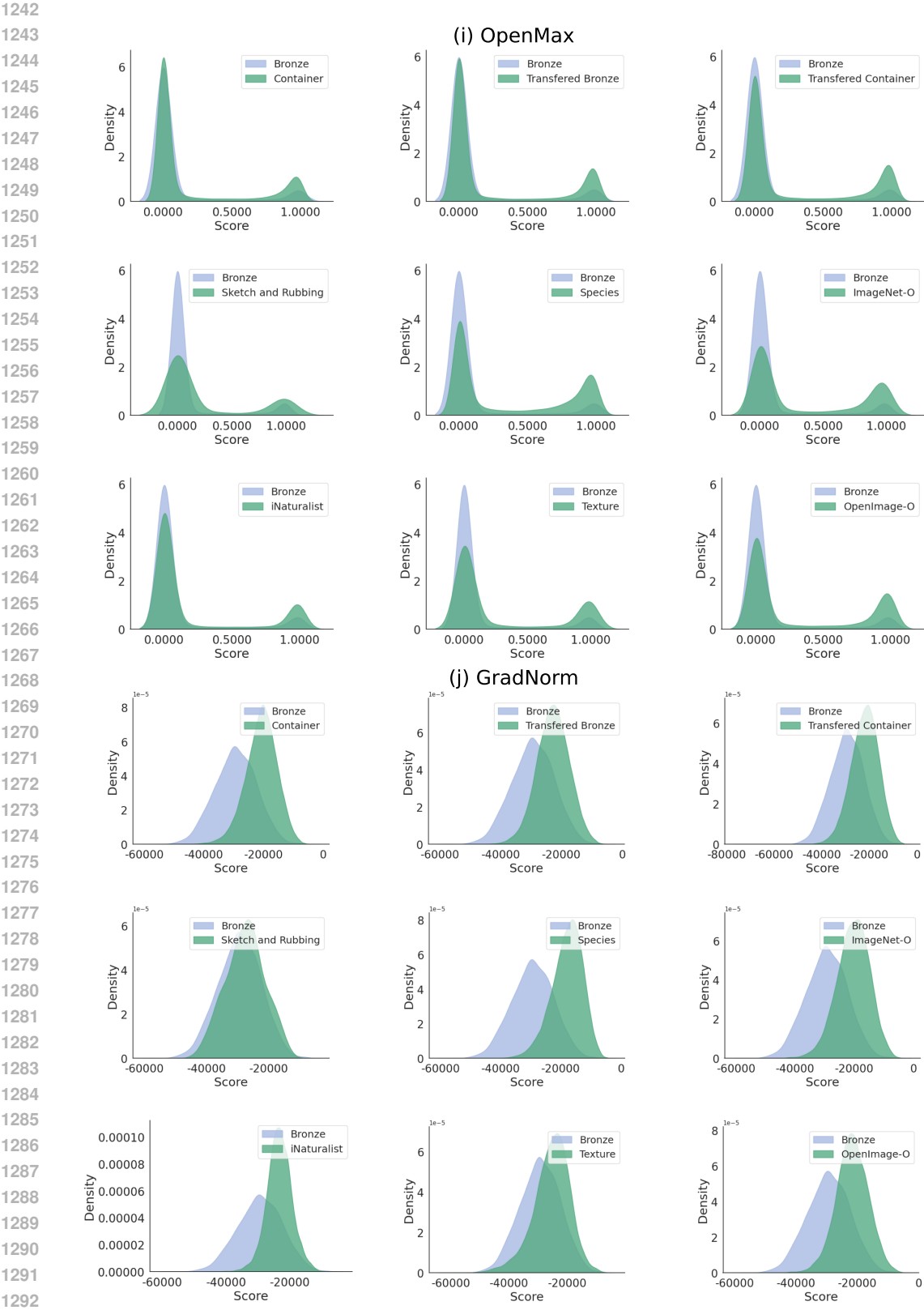

Figure 13: ID and OOD score density distribution of OpenMax (Bendale & Boult, 2016) and Grad-Norm (Huang et al., 2021a) on the ShiftedBronzes and 5 general OOD datasets.

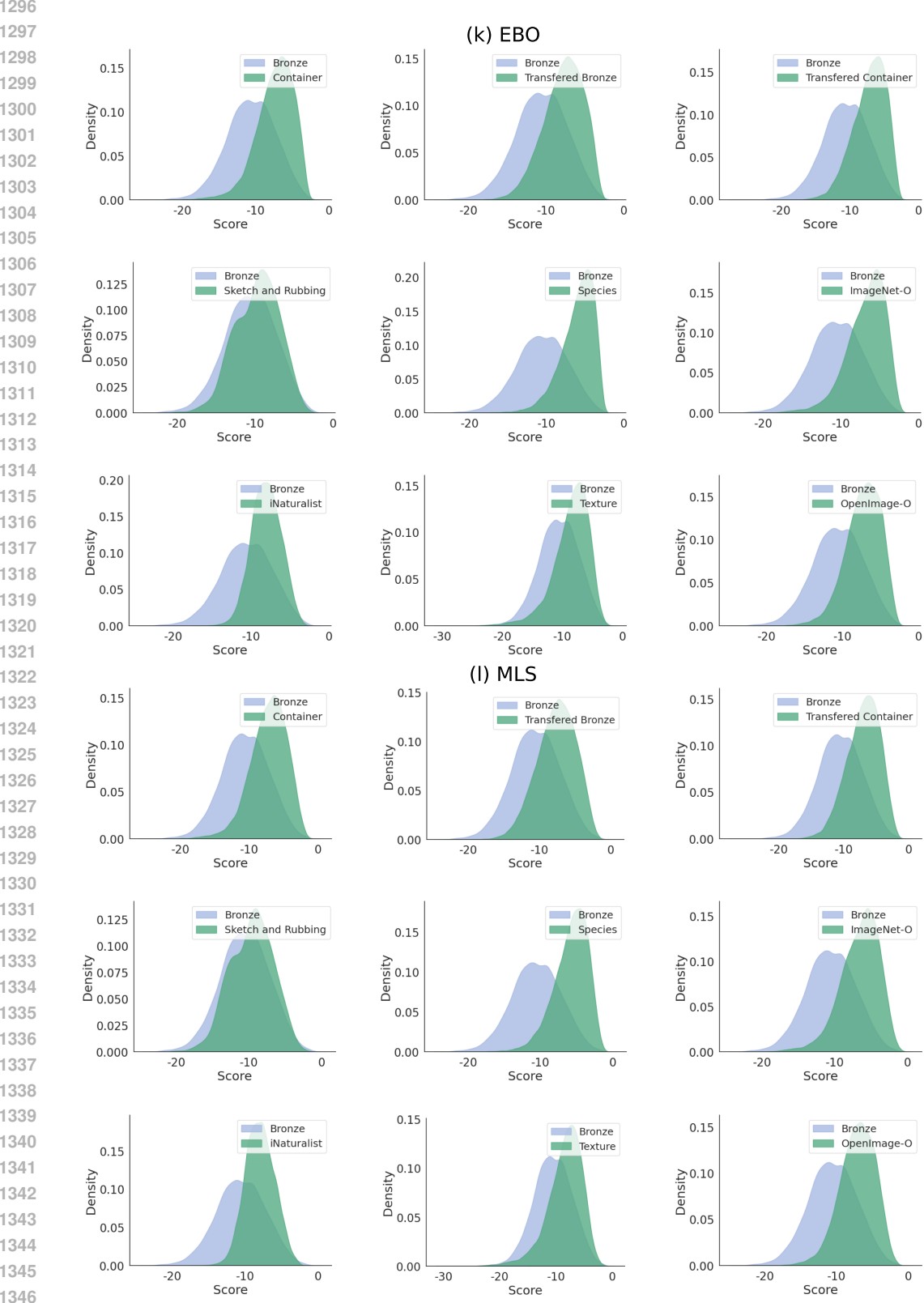

Figure 14: ID and OOD score density distribution of EBO (Liu et al., 2020) and MLS (Basart et al., 2022) on the ShiftedBronzes and 5 general OOD datasets.

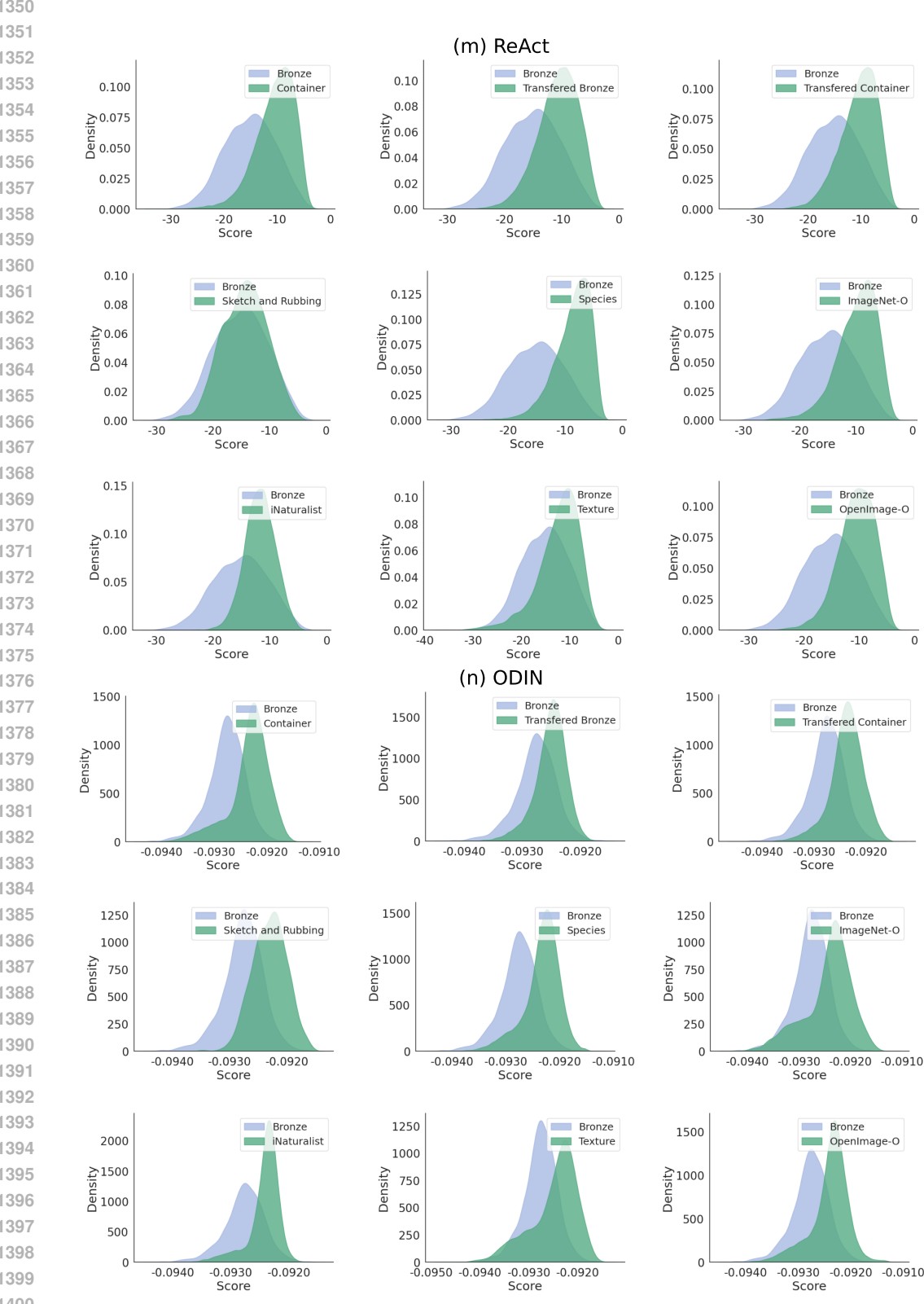

Figure 15: ID and OOD score density distribution of ReAct (Sun et al., 2021) and ODIN (Liang et al., 2018) on the ShiftedBronzes and 5 general OOD datasets.

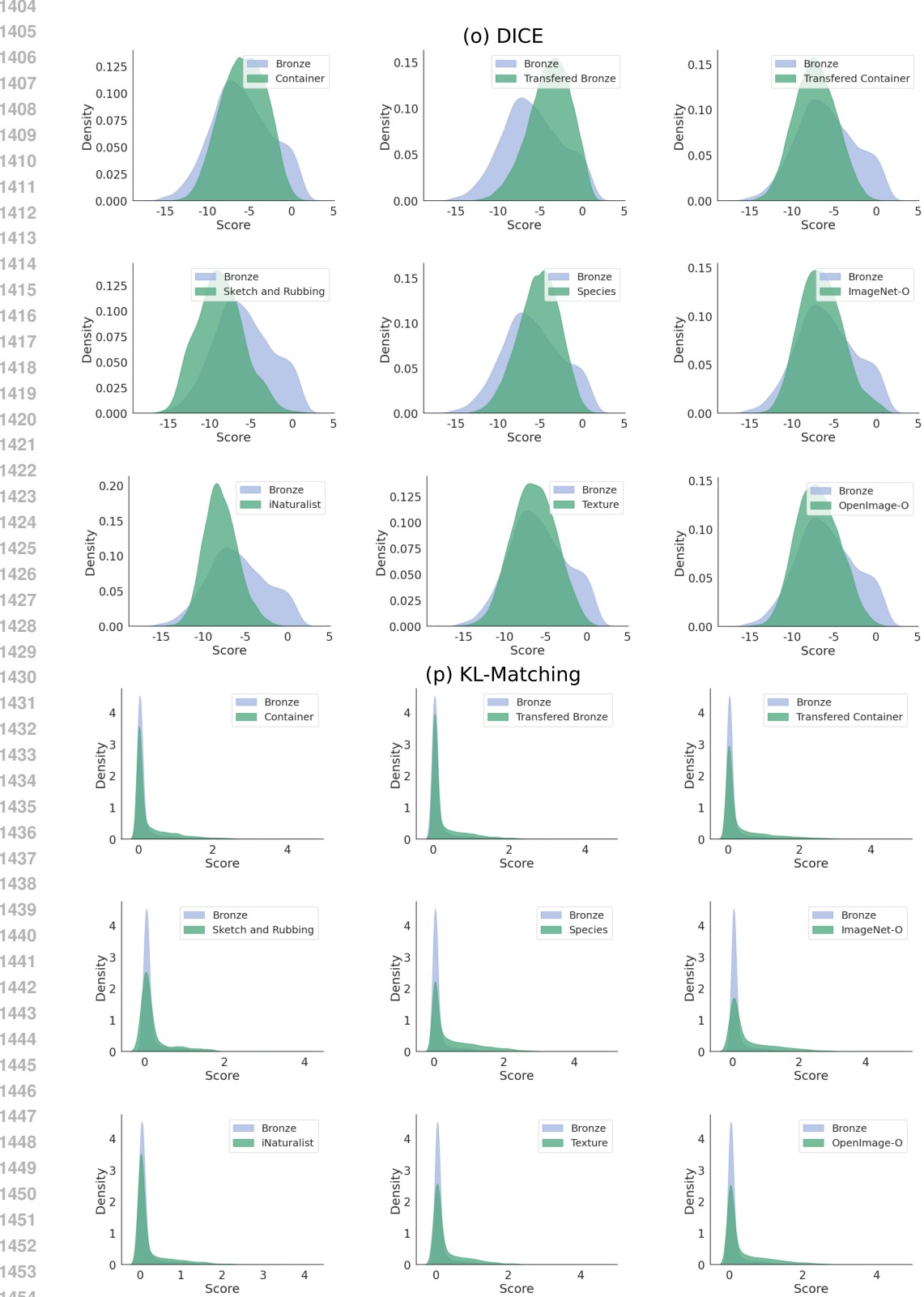

Figure 16: ID and OOD score density distribution of DICE (Sun & Li, 2022) and KL-Matching (Basart et al., 2022) on the ShiftedBronzes and 5 general OOD datasets.

