# OpenReview forum: "ShiftedBronzes: Benchmarking and Analysis of Domain Fine-grained Out-of-Distribution Detection in Gradual Shifts"
_ICLR.cc/2026/Conference — ICLR 2026 Conference Withdrawn Submission_

### Official Review · Reviewer_K4N5 · 2025-10-28

**Soundness:** 2
**Presentation:** 2
**Contribution:** 2
**Rating:** 2
**Confidence:** 4

**Summary:**

This paper tackles the challenge of fine-grained out-of-distribution (OOD) detection, where subtle distribution shifts can significantly degrade model performance. The authors note that existing benchmarks either focus on coarse-grained OOD shifts (e.g., near- vs. far-OOD) or single-level fine-grained shifts, but fail to comprehensively capture the spectrum of nuanced shift levels within fine-grained domains. To address this issue, the paper introduces ShiftedBronzes, a new benchmark specifically designed for fine-grained OOD detection under multiple, systematically controlled shift levels. The authors evaluate a range of post-hoc OOD detectors and vision-language model (VLM)-based methods on ShiftedBronzes as well as several general OOD benchmarks.

**Strengths:**

1. This paper is well-structured and easy to follow.

2. It introduces ShiftedBronzes, the first benchmark specifically designed to evaluate OOD detection under progressively graded and fine-grained distribution shifts.

3. It has been experimentally validated on various OOD detection methods and is supported by comprehensive visualizations.

**Weaknesses:**

1. Regarding the problem setting, I wonder why the paper treats covariate shift data (e.g., background or style variations) as OOD samples, rather than focusing on the model’s generalization performance on these data, such as classification accuracy. This is a more common practice and is central to the field of out-of-distribution (OOD) generalization, since a robust model should be able to maintain reasonably accurate classification performance under various environmental changes, even when the background differs. Therefore, I believe that for datasets such as Sketch and Rubbing, Transferred Bronze, the paper should also report OOD classification accuracy. Many recent studies have jointly considered both OOD generalization and OOD detection, and it would be beneficial for this work to do the same.

    [1] Feed two birds with one scone: Exploiting wild data for both out-of-distribution generalization and detection ICML, 2023.

    [2] Aha: Human-assisted out-of-distribution generalization and detection. NeurIPS, 2024.

    [3] Diversify: A general framework for time series out-of-distribution detection and generalization. TPAMI, 2024

    [4] CRoFT: robust fine-tuning with concurrent optimization for OOD generalization and open-set OOD detection. ICML, 2024

    [5] DeltaEnergy: Optimizing Energy Change During Vision-Language Alignment Improves both OOD Detection and OOD Generalization. NeurIPS, 2025

2. The effects of different types of distribution shifts on OOD detection performance vary. Could you please provid in-depth analysis to explain the underlying reasons for these differences?

3. The experiments are conducted only on the ShiftedBronzes benchmark. I wonder whether the experimental results and conclusions can generalize to other, more general datasets. I believe that varying the in-distribution (ID) datasets could further strengthen the validity and robustness of the findings.

4. The paper does not discuss several recent VLM-based OOD detection methods, such as CSP [6], NegLabel [7], and CLIPScope [8].

    [6] Conjugated semantic pool improves ood detection with pre-trained vision-language models.

    [7] Negative label guided ood detection with pretrained vision-language models.

    [8] Clipscope: Enhancing zero-shot ood detection with bayesian scoring.

5. In Eq 1, Why can the differences in FPR performance across different OOD datasets be used to evaluate a model’s OOD detection capability? What is the theoretical relationship between them?

**Questions:**

See Weaknesses.

---

### Official Review · Reviewer_gdc3 · 2025-10-28

**Soundness:** 2
**Presentation:** 2
**Contribution:** 2
**Rating:** 2
**Confidence:** 4

**Summary:**

The paper presents ShiftedBronzes, a benchmark dataset designed to evaluate the performance of fine-grained classification methods, specifically in the context of bronze ware dating, which is crucial for the study of ancient Chinese history. The dataset expands on the existing bronze Ding dataset by incorporating two types of bronze ware data and seven types of out-of-distribution (OOD) data that exhibit common distribution shifts encountered in bronze ware dating scenarios. The authors conduct benchmarking experiments using various post-hoc, pre-trained VLM-based, and generation-based OOD detection methods on ShiftedBronzes and five general OOD datasets. The analysis of the results validates previous conclusions about these methods while highlighting their distinct behaviors on specialized datasets.

**Strengths:**

1. The paper proposes a interesting benchmark for domain-specific OOD detection, particularly for bronze ware dating, which is a critical task in archaeology.
2. The paper employs a variety of widely adopted OOD detection methods, providing a thorough analysis and comparison of their performance on both domain-specific and general OOD datasets.

**Weaknesses:**

1. The paper would benefit from a more comprehensive summary that contrasts the unique challenges presented by the ShiftedBronzes dataset with those of other domain-specific OOD benchmarks, such as those in medical imaging and drug discovery. The core difficulties inherent in OOD detection within the field of archaeology, particularly in the context of bronze ware dating, are not sufficiently elaborated. Furthermore, the high performance of the ID-like method suggests that the problem may be less challenging than presented or already well-resolved. It is recommended that the authors provide a more nuanced discussion on the complexities and distinctiveness of OOD detection in archaeological applications.
2. The paper lacks a detailed explanation of the practical applications of OOD detection in archaeology. While the identification of counterfeit antiques is posited as a significant application, the selected OOD datasets do not align with this requirement. Specifically, the "Container" dataset does not consist of antique items, and the "Transferred Container" and "Transferred Bronze" datasets are not described with sufficient detail to assess their relevance to the identification of fake antiques. The authors are encouraged to provide a more thorough description and justification of these datasets in relation to the problem of identifying counterfeit archaeological artifacts.
3. The use of ancient bronze dating as a proxy for fine-grained OOD detection is interesting. However, the specific types of shifts created—sketches, rubbings, and AI-driven material transfers (ZeST)—are highly stylistic and artistic in nature. It is unclear if the insights derived from this domain (e.g., VLM's vulnerability to homogenous backgrounds) would directly translate to other fine-grained domains like medical imaging (e.g., different scanners), remote sensing (e.g., seasonal changes), or species classification (e.g., new subspecies). The paper claims the shifts are comparable to these domains but provides no empirical evidence. The benchmark might be testing robustness to a very specific, human-interpretable set of transformations rather than general, subtle distribution shifts.
4. The proposed Distribution-Aware Degree (DAD) metric is defined solely based on the pairwise comparison of FPR95 scores. While FPR95 is a standard OOD metric, it represents only a single point on the ROC curve. A method could have a favorable DAD score by happening to order its FPR95 values correctly, while its overall performance, as measured by AUROC, might be poor or show a different trend. This makes DAD potentially brittle and not fully representative of a method's "distribution awareness." A more robust version of DAD might incorporate multiple metrics (e.g., AUROC, AUPR) or analyze the entire ROC curve's relationship across shift levels. The paper does not justify why FPR95 was the exclusive choice.
5. The Related Work section is largely descriptive rather than analytical. While it enumerates prior benchmarks and methods, it fails to provide a deep, critical synthesis of why existing benchmarks are insufficient for fine-grained analysis or why existing post-hoc methods are theoretically expected to fail on gradual shifts.
6. The paper proposes the DAD metric to measure sensitivity but does not justify why the ability to correctly rank FPR95 scores is a meaningful or sufficient proxy for robust OOD detection.

**Questions:**

N/A

---

### Official Review · Reviewer_7rKQ · 2025-10-29

**Soundness:** 3
**Presentation:** 2
**Contribution:** 3
**Rating:** 6
**Confidence:** 3

**Summary:**

The ShiftedBronzes paper introduces a benchmark for fine-grained out-of-distribution (OOD) detection evaluation under gradual domain shifts. The authors also conduct extensive experiments on both post-hoc OOD detectors and vision-language model (VLM)-based methods. Additionally, they propose a sensitivity metric intended to quantify the robustness of detection models across different shift levels, particularly for VLM-based methods.

**Strengths:**

- The paper presents the first dataset explicitly designed for multi-level fine-grained OOD detection, filling a noticeable gap in current OOD literature.

- The study includes rich experimental comparisons covering multiple categories of OOD detectors, providing valuable empirical insights into how these methods behave under gradual domain shifts.

**Weaknesses:**

- As shown in Table I, the dataset is heavily unbalanced, which could bias both model training and evaluation. Given this is proposed as a benchmark dataset, more balanced design or discussion on mitigating imbalance effects is needed.

- The illustration in Figure 1 is not well-presented. It is difficult to grasp that the two ID bronzes are the main subjects for fine-grained dating categorization. The placement of "Transferred Container" in column (d) is also confusing. It would make more sense if aligned with column (i), as both represent similar shift levels.

- The proposed sensitivity metric's agnosticism claim (model- and dataset-agnostic) is insufficiently demonstrated. No clear ablation or cross-dataset validation is shown to confirm this property.

**Questions:**

- The proposed sensitivity metric is considered to be model- and dataset-agnostic. Which section in this paper demonstrates this statement? Is there an ablation or experiment across different datasets or model families to support this statement? (Figure 6 appears too complicated to interpret this clearly.)

- The set of post-hoc detectors in Table 2 appears dated relative to the current state of OOD detection research. Are there any newer FGVC methods that could strengthen the benchmark's coverage?

- What do the individual blocks in the Figure 2 represent?

---

### Official Review · Reviewer_F17w · 2025-11-01

**Soundness:** 3
**Presentation:** 3
**Contribution:** 3
**Rating:** 4
**Confidence:** 4

**Summary:**

This paper introduces ShiftedBronzes, a benchmark for evaluating fine-grained out-of-distribution (OOD) detection under progressive, multi-level distribution shifts. The authors systematically assess post-hoc detectors and vision-language models, revealing that most methods are insensitive to shift severity and that VLMs tend to overfit to background context. They further propose a Distribution-Aware Degree (DAD) metric to quantify robustness across varying shift intensities.

**Strengths:**

1. This paper target on graded distribution shifts of OOD, and proposes ShiftedBronzes, the first fine-grained OOD benchmark with systematically graded distribution shifts.

2. The authors provide a thorough comparison of post-hoc detectors and VLM-based methods, revealing key failure modes.

3. The authors introduce the Distribution-Aware Degree (DAD) metric, offering a fresh perspective for assessing robustness across shift levels.

**Weaknesses:**

1. All figures in the paper appear to be non-vector images, and some fonts are even blurry, making them hard to read.

2. The colored blocks in Fig.2 are unclear and not explained in the paper. After checking the supplementary materials, the reviewer still could not find any correspondence. Could this issue be caused by missing or unembedded fonts leading to rendering errors in the figure?

3. The evaluation in this paper appears limited and unbalanced. While many post-hoc detectors are included, only three VLM-based methods are evaluated, all relying on CLIP prompt-learning, which has known limitations for OOD detection. It is therefore unclear whether the reported conclusions generalize to more recent approaches. For example, self-supervised features such as those learned by DINO, as well as DETR-based detection frameworks with OOD or domain generalization improvements, have been shown to enhance OOD detection. Similarly, multi-modal LLMs (MLLMs) exhibit strong generalization abilities for detection tasks. Could the authors consider evaluating their benchmark on a broader set of modern methods, such as the Griffon series (Griffon v2, Griffon-G, etc.), to assess whether their experimental conclusions hold across more advanced models?

**Questions:**

See the weaknesses.

---

### Official Review · Reviewer_CSyM · 2025-11-02

**Soundness:** 2
**Presentation:** 2
**Contribution:** 2
**Rating:** 4
**Confidence:** 4

**Summary:**

This paper introduces a dataset of bronze antiquities designed to benchmark out-of-distribution (OOD) detection under distribution shifts. The in-distribution (ID) split contains real images of Gui and Ding artifacts from various historical periods, whereas the OOD split comprises (i) synthetic images of Gui and Ding, (ii) real images of other bronze artifact types, and (iii) objects from unrelated categories (e.g., natural ImageNet classes).

**Strengths:**

- Novel dataset – The collection is unique and will enable systematic, controlled research on OOD detection under distribution shift. Its overall quality represents a solid contribution.

 - Clear writing – The manuscript is well structured and detailed.

**Weaknesses:**

- Experimental mismatch (Table 3) – Most post-hoc OOD detectors evaluated were designed for single-label classifiers, yet they are applied to AKG, which appears to be a multi-label or multi-head model. This mismatch may explain why feature-based methods such as ViM and MDS substantially outperform other post-hoc baselines.

 - Missing recent baselines – Several up-to-date methods (e.g., NNGuide [1], SCALE [2], Forte [3]) were not included. NNGuide, in particular, could suit this fine-grained task if the classifier were single-label. The current pairing of a fine-grained classifier with post-hoc detectors seems sub-optimal, leading to results in which LVLM-based approaches dominate. If re-running the experiments is infeasible, the authors should at least discuss this limitation.

[1] Park, J., Jung, Y. G., & Teoh, A. B. J. “Nearest Neighbor Guidance for Out-of-Distribution Detection.” ICCV, 2023.

[2] Xu, K., et al. “SCALE: Scaling for Training-Time and Post-Hoc Out-of-Distribution Detection Enhancement.” arXiv:2310.00227, 2023.

[3] Ganguly, D., et al. “Forte: Finding Outliers with Representation Typicality Estimation.” arXiv:2410.01322, 2024.

**Questions:**

Please refer to the weaknesses

---

### Note · Authors · 2025-11-14

I have read and agree with the venue's withdrawal policy on behalf of myself and my co-authors.